



# Inferring reservoir filling strategies under limited data availability using hydrological modelling and Earth observation: the case of the Grand Ethiopian Renaissance Dam (GERD)

Awad M. Ali[1,2], Lieke A. Melsen[1], and Adriaan J. Teuling[1]

[1]Hydrology and Quantitative Water Management Group, Department of Environmental Sciences, Wageningen University & Research, P.O. BOX 47, 6700 AA Wageningen, the Netherlands

[2]Water Research Center, Faculty of Engineering, University of Khartoum, P.O. BOX 321, Khartoum, Sudan

**Correspondence:** Awad M. Ali (awad.negmeldinawad.mohammedali@wur.nl)

**Abstract.** The filling of the Grand Ethiopian Renaissance Dam (GERD) started in 2020, posing additional challenges for downstream water management in Sudan, which is already struggling to cope with the effects of climate change. This is also the case for many transboundary rivers that observe a lack of cooperation and transparency during the filling and operation of new dams. Without information about water supply from neighbouring countries, it is risky to manage downstream dams

as usual and operation information is needed to apply modifications. This study aims to test the applicability of using lumped hydrological modelling coupled with remote sensing data in retrieving reservoir filling strategies in regions with limited data availability. Firstly, five rainfall products (namely; ARC2, CHIRPS, ERA5, GPCC, and PERSIANN-CDR) were evaluated against historical measured rainfall at ten stations. Secondly, to account for input uncertainty, the best three performing rainfall products were forced in the conceptual hydrological model HBV-light with potential evapotranspiration and temperature data

from ERA5. The model was calibrated during the period 2006 - 2019 and validated during the period 1991 - 1996. Thirdly, the parameter sets that obtained very good performance (NSE > 0.75) were utilized to predict the inflow of GERD during the operation period (2020 - 2022). Then, from the water balance of GERD, the daily storage was estimated and compared with the storage derived from Landsat observations to evaluate the performance of the selected rainfall products. Finally, three years of GERD filling strategies were retrieved using the best-performing simulation of CHIRPS with RMSE of 1.7 billion cubic

meters (BCM) and NSE of 0.77 when compared with Landsat-derived reservoir storage. It was found that GERD stored 14% of the monthly inflow of July 2020, 41% of July 2021, and 37% and 32% of July and August 2022, respectively. Annually, GERD retained 5.2% and 7.4% of the annual inflow in the first two filling phases and between 12.9% and 13.7% in the third phase. The results also revealed that the retrieval of filling strategies is more influenced by input uncertainty than parameter uncertainty. The retrieved daily change in GERD storage with the measured outflow to Sudan allowed further interpretation

of the downstream impacts of GERD. The findings of this study provide systematic steps to retrieve filling strategies for data-scarce regions, which can serve as a base for future development in the field. Locally, the analysis contributes significantly to the future water management of the Roseires and Sennar dams in Sudan.



# 1 Introduction

The optimal management of water resources is challenging as we increasingly experience the impacts of climate change and
increasing water demand due to population growth (Stakhiv, 2011). Transboundary rivers pose even greater challenges to water
management, for example, when structures such as dams are built by upstream countries, impacting downstream flow regime
(Biswas, 2008) and mean water availability during the filling phase. The absence of collaboration has disastrous impacts in
these regions, as collaboration increases water revenues and reduces the risk of extreme events (Basheer et al., 2018; Wheeler
et al., 2018). There is, however, the possibility of conflicting dynamics in many transboundary basins due to the different
policies and strategies between neighbouring countries (Warner and Zawahri, 2012). As a result, a lack of transparency on the
filling/operation of dams and other infrastructures is often witnessed.

The Blue Nile River is an example of a transboundary river where infrastructure influences downstream dynamics. A current
challenge faced by the Ministry of Irrigation and Water Resources of Sudan (MoIWR) is the management of the Nile River.
Around two-thirds of Nile water is supplied by the Blue Nile River (Dumont, 1986). This river originates in Lake Tana,
Ethiopia, and subsequently flows northward to meet the White Nile River (originates in Lake Victoria, Uganda), where the Main
Nile River starts in Khartoum, the capital of Sudan, as shown in Fig. 1.A. In April 2011, the Ethiopian government started the
construction of the Grand Ethiopian Renaissance Dam (GERD) in the Upper Blue Nile (UBN) basin, around 15 km east of the
Ethiopia-Sudan border. The dam, when completed, will be the largest hydropower dam in Africa with a storage capacity of 74
billion cubic meters (BCM) and a power capacity of 5150 Megawatts (MW) (Ezega News, 2019). It was constructed without an
agreement with downstream countries (i.e., Sudan and Egypt), which caused hydro-political tension between Sudan, Egypt, and
Ethiopia (Gebrehiwet, 2020). Moreover, water management in Sudan is very closely tied to reservoirs. The Roseires dam, for
example, plays a significant role in managing drinking water as well as irrigation water for downstream projects and is closely
coordinated with the Sennar dam downstream (Alrajoula et al., 2016) (see Fig. 1.B). Around 35% of the Sudan allocation of
Nile water is consumed by the Gezira scheme, which is supplied by the Sennar dam (Adam et al., 2003). The Gezira scheme,
with a total area of 8800 km$^2$, is regarded as one of the largest irrigation schemes in the world under a single management
structure (Ahmed et al., 2006). It, therefore, plays a considerable role in the socio-economy of Sudan.

The construction of GERD, in combination with a low level of coordination between Ethiopia and Sudan, poses a threat to
water management in Sudan. For instance, floods are frequently experienced in Khartoum state, where 16% of the population
lives (World Population Review, 2022). In the past 35 years, flood events were observed, on average, every 5 years. The latest
flood of 2020 was unprecedented, where water levels exceeded all previous events (NASA Earth Observatory, 2020). In the
same year, the filling of GERD started without sharing information with Sudan and Egypt. As shown in Fig. 1.B, the Roseires
dam is located only around 120 km downstream of GERD. Without knowledge of GERD operation strategies, management of
the Roseires dam is a considerable challenge. Knowledge of operation strategies is thus essential for flow prediction and water
management in downstream areas.

It is clear that the operation of the dams along the Blue Nile River has significant implications for food security, flood pro-
tection, water availability, and hydropower generation downstream in the Main Nile River. Nonetheless, Ethiopia has yet to





share information on GERD filling, thus hindering water management in Sudan. Therefore, obtaining filling/operation information using alternative means is urgently needed to support the operation of both; the Roseires and Sennar dams. Different approaches exist to understand dam operation and reservoir water storage. One approach relies on satellite remote sensing and

radar altimetry. At a global scale, satellite-derived water heights and extents were used to reconstruct storage dynamics (Hou et al., 2022). Vu et al. (2022) revealed reservoir filling strategies and operating rules of ten dams along the Upper Mekong River using Landsat images & radar altimetry. Another approach is using a process-based hydrological model to simulate water inflow, storage, and release. Eldardiry and Hossain (2019) predicted reservoir operating rules of High Aswan Dam (HAD) in the lower Nile River basin using satellite hydrometeorological observation and a macroscale hydrologic model. Also, Wannasin

et al. (2021) simulated the daily storages of two reservoirs using a distributed hydrological model and a reservoir operation module. In this study, the quantification will be based on the latter approach for three main reasons: (1) Roseires dam is very close to GERD, thus hydrological modelling can support proactive flow prediction for real-life operations (2) Finer and more consistent time steps can be achieved when using hydrological models (3) The impact of rainfall uncertainty on inferring filling strategies can be tested. However, the first approach will be utilized in the validation stage as will be detailed later.

This study aims to test the applicability of lumped hydrological modelling in large basins to retrieve reservoir-filling strategies with limited data availability. As part of this study, we will analyze the implications of input data and parameter uncertainty in inferring the filling strategies. As a case study, the research will be conducted on the UBN basin to understand the filling stages of GERD and its impact on downstream discharge. Specifically, the study will quantify the volume of water stored behind the GERD based on known outgoing flow to Sudan as well as the daily reservoir storage change. The output of this

study intends to provide information to support the management of the Blue Nile River in Sudan. The outline of the study is as follows: the study area and data are briefly described in Section 2. The methodology, including the evaluation of selected rainfall products, the use of hydrological modelling, and the inference of reservoir storage, are detailed in Section 3. In Section 4, the results of the study are presented and discussed in Section 5. Finally, the conclusion of the work done is summarized in Section 6.

## 2  Study area and data


### 2.1  Upper Blue Nile basin

The Blue Nile River basin has a drainage area of 310,000 km$^2$ that is shared between Ethiopia (64%) and Sudan (36%). From lake Tana, in the north-western Ethiopian Highlands, at 1780 m a.s.l the water is supplied and flows clockwise through the eastern mountainous side of the basin to the western lower altitude side. A few kilometres after the Ethiopia-Sudan border, the

water is measured at the outlet of the Upper Blue Nile (UBN) basin at Eldiem station (481 m a.s.l). Afterward, the water is drained from the Lower Blue Nile (LBN) basin through the Blue Nile River and its tributaries; the Dinder River and the Rahad River, to the Main Nile River as shown in Fig. 1.B.

The UBN basin will be the area of interest for this study which covers an area of around 176,000 km$^2$. The basin is subdivided into 14 sub-basins as shown in Fig. 1.C. The rainfall of the UBN basin ranges from 1000 mm yr$^{-1}$ near the Ethiopia-Sudan



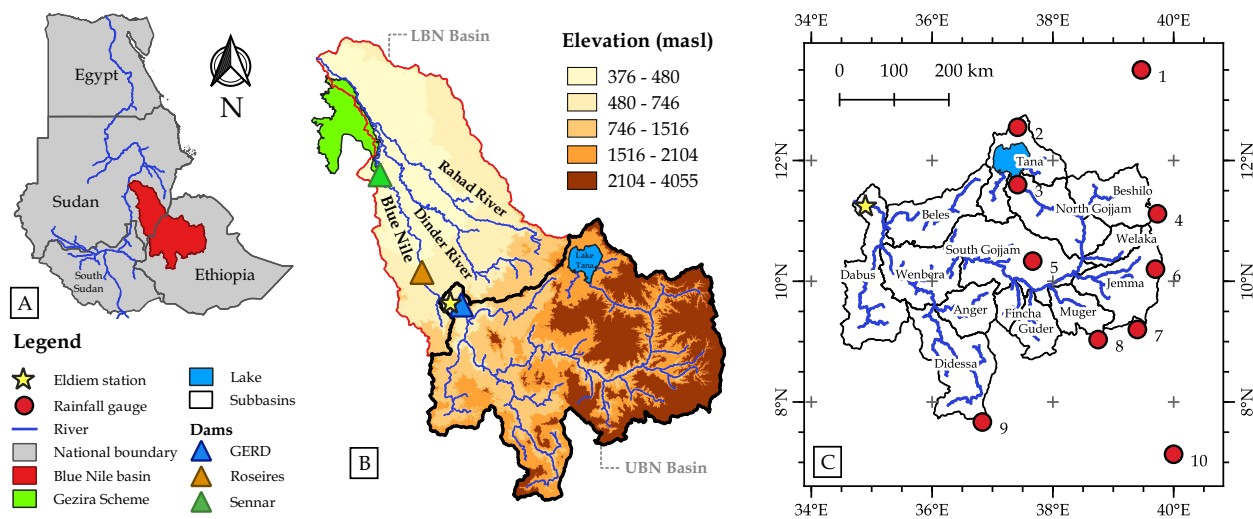

**Figure 1.** Study area overview: (A) The geographical location of the Blue Nile Basin with the Upper and Lower parts, and (B) a Digital Elevation Model (SRTM 30m) map of the Blue Nile Basin and location of the Gezira Irrigation Scheme, dams and the Eldiem streamflow station, (C) Upper Blue Nile subbasins with the location of the rainfall gauge stations. Basins boundaries and drainage network are obtained from the HydroSHEDS dataset (Lehner et al., 2008) and national boundaries were based on the Database of Global Administrative Areas (GADM, 2012). The name and coordinates of each rainfall gauge station are demonstrated in Table 3.

border to 2200 mm yr$^{-1}$ in the Didessa and Dabus subbasins, reflecting the high spatial variability of precipitation associated with topographical features (i.e., so-called water towers) that is typical for much of Eastern Africa (Wamucii et al., 2021). There is a high, but spatially variable, mean potential evapotranspiration, which varies between 1000 and 1800 mm yr$^{-1}$ (Conway, 2000). A typical annual air temperature in the basin ranges from 13°C to 26°C (Tekleab et al., 2013). The discharge regime of the Blue Nile River is highly seasonal, with more than 80% of its annual discharge occurring between July and October and

only 4% between January and April (Kim and Kaluarachchi, 2009; Sutcliffe et al., 1999).

    Four main land uses dominate in the UBN basin, which are cropland (54.74%), open forest (23.62%), shrubland (8.29%), and closed forest (7.94%) while, the remaining land uses cover only 5.41% of the total area based on the Copernicus Global Land Service (CGLS) (Buchhorn et al., 2020). The large basin is composed of heterogeneous soil types. According to Soil-GRIDS dataset (Hengl et al., 2017), the most common types are Leptosols, Luvisols, Vertisols, and Nitisols, respectively, which

collectively occupy 96% of the area.

## 2.2   Ground measurements

The scarcity of ground measurements in terms of availability and accessibility is a challenge and motivation for this study. Streamflow stations are scattered in the basin and cover many subbasins, but they are not publicly available. Downstream GERD, and after the border, Sudan measures the incoming water at Eldiem station since the 1960s (Fig. 1.B). This data was



**Table 1.** Main characteristics of the available ground measurements.

| Data Type | Temporal Resolution | No. of Stations | Available Years |
|---|---|---|---|
| Discharge | Daily | 1 | 1990 - 2022 |
| Rainfall | Monthly | 8 | 1984 - 2005 |
| Rainfall | Monthly | 2 (St. 6&7)* | 1993 - 1999 |

\* See Fig. 1.C

obtained from the MoIWR for the period 1990-2022 with a daily timestep. Consequently, the calibration and validation of the hydrological model will be based on these measurements as will be detailed later. It is worth mentioning that the discharge at Eldiem station is estimated from measured water levels using rating curves. From 2012 onwards, the Eldiem Station is influenced by the backwater effect of the Roseires dam during filling months (June - October). During these months, the MoIWR estimates the discharge from the water balance of the Roseires dam.

Despite the availability of more than 1200 meteorological stations in Ethiopia, the existing stations within the UBN basin are not sufficient to cover the spatial variability of the rainfall. Additionally, the measured rainfall data is not freely available from the National Meteorological Agency (NMA) of Ethiopia. However, historical monthly rainfall measurements at ten stations were obtained from the NMA as indicated in Fig. 1.C and Table 1. Therefore, rainfall measurements will be used to validate satellite and reanalysis rainfall products available for this region, as remote sensing products cover the entire basin and recent

data are available.

## 2.3 Input data

Due to the unavailability of the required observed input data, remote-sensing products will be used in this study. These data include meteorological forcing data (precipitation, potential evapotranspiration, and temperature), and satellite imagery. The spatial resolution and references of the selected datasets are summarized in Table 2 and will be utilized for different purposes

which are detailed in the following sections.

In this study, the quality of five rainfall products will be tested including the African Rainfall Climatology version 2 (ARC2), the Climate Hazards Group InfraRed Precipitation with Station Version 2 (CHIRPS), the fifth-generation ECMWF atmospheric reanalysis (ERA5), the Global Precipitation Climatology Centre full data reanalysis version 7 (GPCC), and Precipitation Estimation from Remotely Sensed Information using Artificial Neural Networks - Climate Data Record (PERSIANN-CDR). The

potential evaporation and temperature data are retrieved from ERA5.





**Table 2.** Summary of the remote sensing products used as inputs in the hydrological modelling over the Upper Blue Nile (UBN) basin.

| Input | Dataset | Spatial resolution | Reference |
|---|---|---|---|
| Satellite imagery | Landsat | 30 m | (Survey, 2022) |
| Potential Evapotranspiration | ERA5 | 0.25° | (Hersbach et al., 2019) |
| Temperature | ERA5 | 0.25° | (Hersbach et al., 2019) |
| Precipitation | ARC 2.0 | 0.1° | (Novella and Thiaw, 2013) |
| | CHIRPS | 0.05° | (Funk et al., 2014) |
| | ERA5 | 0.25° | (Hersbach et al., 2019) |
| | GPCC | 1.0° | (Ziese et al., 2020) |
| | PERSIANN-CDR | 0.25° | (Ashouri et al., 2015) |

## 3 Methodology

### 3.1 Evaluation of selected rainfall products

As rainfall products will be used as forcing data, it is important to understand how these data perform in comparison with the available ground measurements. The selection of a representative product is likely to be crucial in minimizing model output uncertainty. In this step, the available monthly rainfall observations in the sites shown in Fig. 1.C were used to investigate the performance of the five mentioned rainfall products using a point-to-pixel approach (see Fig. 2). The approach compares the time series of observed rainfall to the gridded products assuming that grid-cell values are represented by the corresponding rain gauge measurements (e.g., Thiemig et al., 2012; Basheer and Elagib, 2019; McNamara et al., 2021). This approach is useful for data-scarce regions.

A statistical validation using six performance metrics (three linear fit metrics: $R^2$, Intercept, and Slope; and three error metrics: MAE, MBE, and RMSE) were calculated using the equations in Table A1 in the appendices. Additionally, to draw general conclusions, the products were ranked based on the calculation of the Unified Metric (UM: Eq.(1)) and the Overall Unified Metric (OUM: Eq.(2)) (Elagib and Mansell, 2000). The UM indicates the ranking of each product at each station, while the OUM is the sum of the UMs for each product at all stations. Firstly, each product is ranked per station for each metric. Secondly, the performance rankings are summed up per product with a low value indicating good performance and a high value indicating poor performance relative to each other.

$$\text{UM}_{p,i} = \sum_{m=1}^{K} R_{p,m,i} \tag{1}$$

$$\text{OUM}_p = \sum_{i=1}^{L} \text{UM}_{p,i} \tag{2}$$

where $R_{p,m,i}$ is the rank of the product $p$ based on the performance metric $m$ at station $i$, and $\text{OUM}_p$ is the Overall Unified Metric of the rainfall product $p$. While $K$ and $L$ are the number of considered performance metrics and rainfall stations,





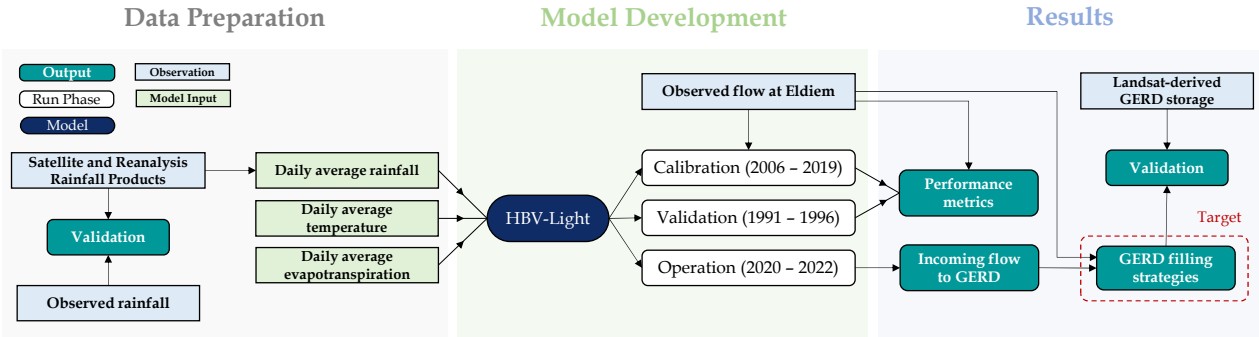

**Figure 2.** Flowchart representing research methodology. Steps are shown in chronological order, the execution is carried out from left to right. For rainfall product validation five products are selected which are ARC 2.0, CHIRPS 2.0, ERA5, GPCC, and PERSIANN-CDR. For storage validation, observed storage will be computed by converting the water surface area observed by Landsat to water volume using an Elevation-Area-Storage relationship of the GERD reservoir.

respectively. UM ranges from $K = 6$ to $K \times$ No. of rainfall products = 30. While the minimum OUM value is 60 ($K \times L$) and the maximum value is 300 ($K \times L \times$ No. of rainfall products).

Based on this analysis, the first three ranked rainfall products will be selected to analyse the influence of the uncertainty in the rainfall product on retrieving reservoir storage strategies. This is due to the fact that precipitation changes can dominate
reservoir volume changes (Hou et al., 2022). It is important to note that the rank of the rainfall products might differ if the analysis was based on the daily timestep that is used in our hydrological model simulations.

### 3.2  Hydrological modelling

From 57 previous studies in the literature between 2007 and 2022, 26 hydrological models have been implemented in the Upper Blue Nile basin (see Table B1). The Soil and Water Assessment Tool (SWAT) and the Hydrologiska Byråns Vattenbal-
ansavdelning (HBV) are the most commonly used models, utilized in 23 and 13 studies, respectively. Both models performed well in simulating the runoff at the outlet of the UBN basin (Betrie et al., 2009; Teklesadik et al., 2017). Bizuneh et al. (2021) compared their performance in the UBN basin and concluded a better performance of SWAT in the three studied watersheds. However, the HBV conceptual model will be used due to its lower demand for input data compared to the SWAT model and the limited number of parameters. HBV is a widely used conceptual hydrological model developed by Bergström et al. (1995), but
with several different implementations being used (Jansen et al., 2021). In this study, a lumped representation of the original HBV model will be used which is HBV-light (Seibert and Vis, 2012). The lumped model simulates daily discharges using daily average precipitation, temperature, and potential evapotranspiration over the basin as input (Seibert, 1996). In addition to its flexibility and computational efficiency, HBV has been successfully applied in the UBN basin (e.g., Uhlenbrook et al., 2010).

HBV-light will be calibrated for a period of 14 years (70%) from 2006 to 2019 and afterward validated for 6 years (30%)
from 1991 to 1996 (see Fig. 2). The selection of the years was due to a considerable gap in the measurements from 1997





to 2000. As a warm-up year, one year before each period (i.e., 2005 and 1990) will be used to avoid the effect of the initial values. Furthermore, to account for parameter uncertainty, the generalized likelihood uncertainty estimation (GLUE) method will be adopted (Beven and Freer, 2001). The GLUE method was employed as it is simple and commonly used in hydrology. In hydrological models, Nash-Sutcliffe Efficiency (NSE) proposed by Nash and Sutcliffe (1970) is widely used as the objective

function (Setegn et al., 2010). Therefore, different parameter sets will be selected based on NSE exceeding a threshold value of 0.75 which is classified as very good performance according to Moriasi et al. (2007). Moreover, to evaluate the performance of the hydrological model, Moriasi et al. (2007) classification will be used.

### 3.3 Retrieving GERD filling strategies

#### 3.3.1 Reservoir area and storage from satellite observations

As observations of GERD filling are currently not available, the quantification of the reservoir storage using Landsat images will allow to validate the ability of the hydrological modelling in retrieving the filling strategies (see Fig. 2). This can also provide insights into the sensitivity of storage estimation to the selection of rainfall products.

To derive reservoir storage from Landsat images, Landsat Collection 2 Level 2 was obtained. Landsat was selected due to its high spatial resolution (30 m) as well as its reasonable revisit time (16 days). Additionally, to increase the temporal resolution

up to 8 days, images from multiple satellites (i.e., Landsat 7/8/9) were used following the steps shown in Fig. 6.A. Different indices allow the estimation of the Water Surface Area (WSA), in this study the Normalized Difference Water Index (NDWI) based on Eq.(3) was selected. Hence, three bands are needed from Landsat which is: the Green (GRN), Near Infrared (NIR), and Quality Assessment bands.

$$\text{NDWI} = \frac{\text{GRN} - \text{NIR}}{\text{GRN} + \text{NIR}} \tag{3}$$

Firstly, the images over the GERD reservoir were obtained from the three satellites. However, the images of Landsat 7 contain missing data stripes caused by the Scan Line Corrector (SLC) failure (Scaramuzza and Barsi, 2005). The missing data were filled using the Fill nodata tool of the Quantum GIS (QGIS, 2009). Then, as the GERD reservoir is covered by two tiles, the tiles were mosaicked and clipped over the reservoir extent. Based on the NDWI maps, the algorithm of Vu et al. (2022) use the data to estimate the WSA. This algorithm is an improvement of the algorithm introduced by Gao et al. (2012) and modified by

Zhang et al. (2014) to allow for the use of Landsat Collection 1 data. The algorithm improves the estimation of the images with clouds using the cloudless images by removing the clouds, clouds' shadows, and no-data pixels. The improvement is two-fold; the selection of cloudless images and the identification of additional water zones. From 2022 onwards, Collection 2 structures only will be used for new Landsat data. Therefore, the algorithm was adapted for the processing of all new Landsat acquisitions. Afterward, to convert the area to storage, an Area-Storage relationship is required. For this, the Elevation-Area-Storage

relationship of the reservoir was produced from the Shuttle Radar Topography Mission (SRTM) digital elevation model (Farr et al., 2007) using the python code written by Vu et al. (2022). Finally, dates with suspicious values were checked manually and erroneous values were removed. The unrealistic values were found to occur mainly when the images are completely covered with clouds.





### 3.3.2 Reservoir storage from hydrological modelling

Since July 2020, GERD started filling and the assumption that the observed flow to Sudan (at Eldiem) is equal to the incoming flow to GERD is no longer valid. Accordingly, to predict the unknown inflow of GERD after 2020, the calibrated hydrological models will be utilized. HBV-light will be run for the period 2020-2022 using the optimized parameter sets (see Fig. 2). Thereafter, the observed and simulated discharge together will be used to estimate the water storage volume at the reservoir using the following equations:

$$Q_{out} = Q_{in} - \frac{dS}{dt} - E \tag{4}$$

$$\frac{dS}{dt} = \theta \cdot Q_{in} - E \tag{5}$$

$$\theta = 1 - \frac{Q_{out}}{Q_{in}} \tag{6}$$

where $Q_{in}$ and $Q_{out}$ are the daily inflow to and outflow from the dam, respectively, $\frac{dS}{dt}$ is the daily change in reservoir storage volume, $E$ is the evaporated volume of water from the reservoir, and $\theta$ is the fraction of inflow volume retained by the

reservoir which ranges between 0 and 1. The last two equations (i.e., Eq. 5 and 6) is used to estimate the daily storage which when accumulated gives the total stored water in the reservoir. The daily inflow $Q_{in}$ will be estimated from the hydrological modelling, while the daily outflow $Q_{out}$ will be based on discharge measurements at Eldiem station. To calculate $E$ for each timestep, the estimated daily average evaporation per month obtained from Khairy et al. (2019) is multiplied by the reservoir surface area at that timestep. The area is estimated from the storage using the acquired Area-Storage relationship.

## 4 Results

### 4.1 Performance of the rainfall products

The performance of the five selected rainfall products based on ten rain gauge stations was estimated and illustrated in a boxplot for each performance metric as shown in Fig. 3. From the boxplots, it can be revealed that CHIRPS has the most accurate estimates as it is the closest product to the optimal values (horizontal dashed red lines) in most of the evaluated

metrics. CHIRPS has a high mean $R^2$ of 0.89, a mean slope of 0.96, and a relatively low mean intercept of 10 mm. CHIRPS has the lowest values for the error metrics which were found to be 36.66 mm for the RMSE, 5.31 mm for the MBE, and 23.7 mm for the MAE. Moreover, GPCC and ARC2 exhibited the best mean slope and intercept, respectively. On the other hand, ERA5 shows the highest overestimation (mean slope of 1.28) and the highest error values. Additionally, it is also clear that ARC2 consistently underestimates the rainfall in all locations (negative MBE values). The performance of the rainfall products

also varied spatially at the different stations. The spatial distribution of the calculated metrics (see Fig. A1) demonstrated consistent performance of CHIRPS over the spatial extent and high spatial variation of ERA5. For further demonstration of the metrics' values and spatial distribution see Fig. A1 and A2 in the appendices.





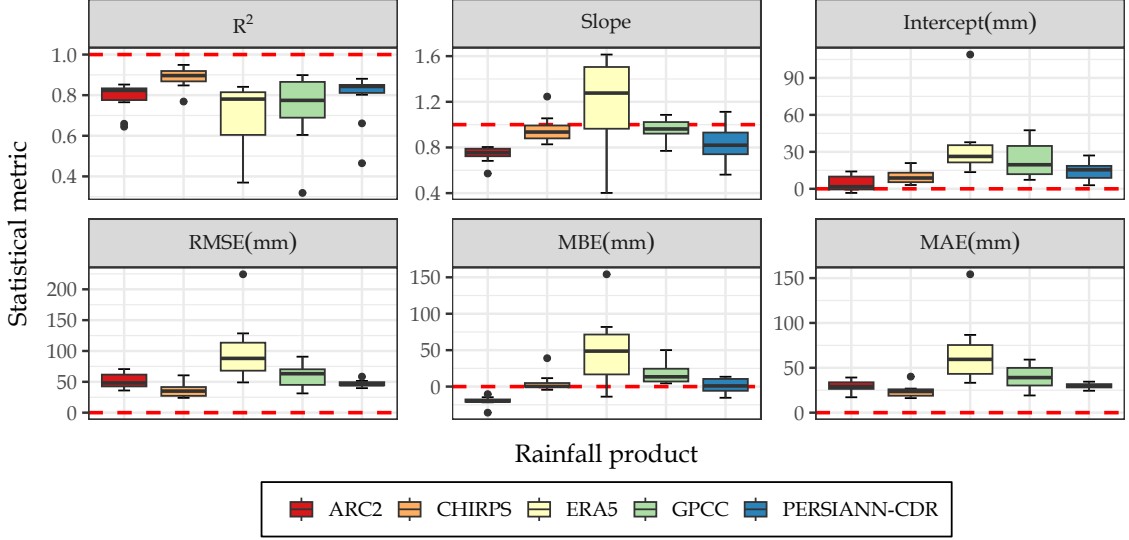

**Figure 3.** Performance of the five selected rainfall products. Each box reflects the distribution over ten stations (see Fig. 1.C and Table 3). The horizontal dashed red lines indicate the optimal values. Detailed calculations are illustrated in Fig. A2 and the equations are listed in Table A1.

Furthermore, the UM in Table 3 emphasizes the accuracy of CHIRPS (i.e., has the lowest values) almost in all stations, unlike ERA5 which was the least accurate product. Additionally, the calculation of the OUM concluded that the overall best-performing product is CHIRPS followed by PERSIANN-CDR, ARC2, GPCC, and ERA5. Therefore, CHIRPS is expected to perform better in simulating the discharge in the river, estimating the water stored behind the dam, and retrieving the filling strategies. However, the first three rainfall products were selected as forcing data for the hydrological modelling to account for the input uncertainty in retrieving the filling strategies.

## 4.2 Hydrological modelling using HBV-light

### 4.2.1 Model calibration and validation

The HBV-light model was forced by ERA5 temperature and potential evapotranspiration estimates while CHIRPS, PERSIANN-CDR, and ARC2 rainfall data were provided to account for input uncertainty. Initially, the parameter ranges tested in the Gilgel Abay catchment in the UBN basin by Uhlenbrook et al. (2010) were adopted. However, the model performance was limited by the initial range of the maximum soil moisture (FC) which was 200 to 600 mm. Therefore, a new range was selected from 200 to 1000 mm which was sufficient to tackle this issue. HBV-light routines, parameters symbol and description, and parameter ranges are listed in Table C1.

To account for parameter uncertainty, 10,000 random parameter sets were generated by the Monte Carlo runs of HBV-light based on the parameter ranges and assuming a uniform distribution. After running the model during the calibration period, 1756





**Table 3.** Ranking of the Satellite and Reanalysis Rainfall Products for each station based on ten selected rain gauge stations. UM is the Unified Metric and OUM is the Overall Unified Metric.

| No. | Station | Longitude | Latitude | Unified Metric (UM) | | | | |
|---|---|---|---|---|---|---|---|---|
| | | | | CHIRPS | GPCC | ERA5 | PERSIANN-CDR | ARC 2.0 |
| 1 | Mekelle | 39.5 | 13.5 | 7 | 16 | 25 | 25 | 17 |
| 2 | Gonder Airport | 37.4 | 12.6 | 11 | 8 | 27 | 20 | 24 |
| 3 | B-dar Synoptic | 37.4 | 11.6 | 8 | 25 | 18 | 15 | 24 |
| 4 | Combolcha | 39.7 | 11.1 | 9 | 19 | 28 | 16 | 18 |
| 5 | Debre Markos | 37.7 | 10.3 | 7 | 23 | 29 | 15 | 16 |
| 6 | Mehal Meda | 39.7 | 10.2 | 14 | 19 | 30 | 13 | 14 |
| 7 | Shola Gebeya | 39.4 | 9.2 | 8 | 15 | 29 | 16 | 22 |
| 8 | Addis Ababa Bole | 38.8 | 9.0 | 9 | 23 | 29 | 15 | 14 |
| 9 | Jimma | 36.8 | 7.7 | 8 | 27 | 20 | 13 | 22 |
| 10 | Robe | 40.0 | 7.1 | 11 | 24 | 26 | 15 | 14 |
| | OUM | | | 92 | 199 | 261 | 163 | 185 |
| | Overall rank | | | 1 | 4 | 5 | 2 | 3 |

model simulations achieved NSE > 0.75 when using CHIRPS while only 269 and 244 simulations in the case of PERSIANN-
CDR and ARC2, respectively. Fig. C1 shows the range of each parameter for the very good simulations in comparison to the applied parameter range (with narrower ranges indicating sensitivity). As a result, the simulated discharge was found to be sensitive to FC, BETA, K1, and K2 when using the three rainfall products.

Fig. 4.A and 4.B shows the daily time series of simulated, using CHIRPS, and observed discharges over the calibration and validation periods, respectively (see Fig. C2.A and C2.B for both PERSIANN-CDR and ARC2). The parameter set that
obtained the highest performance for each rainfall product is indicated in Table C1. Looking at the best simulation, it can be revealed that HBV-light was able to capture the intra-annual seasonality in the three rainfall cases and in both periods but hardly capture the daily variation in discharge, especially in high flows which were also remarked by Uhlenbrook et al. (2010). Accordingly, the models resulted in very good NSE values (as shown in Table 4) and dropped after removing the seasonality to satisfactory (0.51) and unsatisfactory (-0.32 and -0.25) for CHIRPS, PERSIANN-CDR and ARC2 during calibration years,
respectively.

Table 4 also allows comparing the behaviour of the model when utilizing the different rainfall products. Firstly, the performance of the best simulations was represented by the first three metrics (NSE, PBIAS, and RSR) with their equations in Table A1. For all rainfall products, NSE and RSR was found to be very good in the calibration and validation runs except for PERSIANN-CDR during validation which were found to be good, confirming the skilful predictability of the HBV-light model.
On the other hand, PBIAS was generally very good but ARC2 and PERSIANN-CDR overestimated the discharge when applied from 1991 to 1996, resulting in satisfactory and unsatisfactory performance, respectively. Secondly, the P-Factor and R-Factor





were used to measure the uncertainty. The highest percentage of the observations bracketed by the 95PPU was in the case of CHIRPS (78% and 85%) whereas the lowest percentages were ARC2 during calibration (38%) and PERSIANN-CDR during validation (49%). Additionally, the uncertainty range of CHIRPS was found to be the widest (0.73 and 0.78) and ARC2 the

narrowest (0.35 and 0.39). Overall, from the analysis of model performance and uncertainty, it can be concluded that CHIRPS is the best product for simulating the discharge at the outlet of the UBN basin followed by ARC2 and then PERSIANN-CDR.

### 4.2.2 Predicted inflow to GERD

Since 2020, the inflow to the GERD dam remains unknown to Sudan due to data sharing challenges. Therefore, the main purpose of the HBV-light model in this study was the predictability of the inflow to GERD. The selected parameter sets in the

last step were run from 2020 to 2022. Fig. 4.C illustrates the daily inflow (from HBV-light using CHIRPS) and outflow (from Eldiem station) of GERD (see Fig. C2.C for both PERSIANN-CDR and ARC2). It is important to note that PERSIANN-CDR data were available up to June 2022 (before the third filling phase).

Firstly, for the three rainfall products, it is obvious that during filling dates (vertical grey shaded areas) the inflows were greater than the outflow indicating the occurrence of reservoir filling. However, the filling periods were better captured by

CHIRPS followed by PERSIANN-CDR and ARC2 which gave longer filling periods (longer periods where inflow > outflow). Moreover, PERSIANN-CDR demonstrates two filling phases in July and August of the first year.

Secondly, during no-fill dates, water passes through the dam so it is expected that the inflow is equal to the outflow. However, due to model, parameter, and data uncertainties, this is difficult thus outflow is supposed to be within the uncertainty range. Only for CHIRPS runs, the 95PPU range was able to bracket the discharge during no-fill dates. In the case of PERSIANN-

CDR, the inflow was significantly lower than the outflow, falsely indicating a release from the reservoir (no emptying of the reservoir occurred during this period). Lastly, ARC2 showed very high inflows from June to October suggesting a large storage amount.

Overall, based on the analysis during the filling period, CHIRPS was efficient in reproducing the hydrological properties of the system after 2020, while PERSIANN-CDR and ARC2 missed important features (i.e., filling dates).

## 4.3 Reservoir storage estimation

### 4.3.1 Elevation-Area-Storage (E-A-S) relationship

The Elevation-Area-Storage (E-A-S) relationship is essential in this work for two reasons; (1) converting the WSAs observed by Landsat to storages and (2) estimating the WSA corresponding to daily modelled GERD storage hence calculating the total daily evaporation from the reservoir. Therefore, the E-A-S relationship was derived from the SRTM starting with estimating

the area corresponding to each water level (E-A relationship) at 1 m increments. Next, the reservoir volume corresponding to each water level was calculated using the trapezoidal approximation (Gao et al., 2012; Bonnema and Hossain, 2019). The obtained E-A, A-S, and E-S relationships were fitted with a fifth-degree polynomial as illustrated in Fig. 5. Further detailed information about the approximation can be found in the study by Vu et al. (2022).





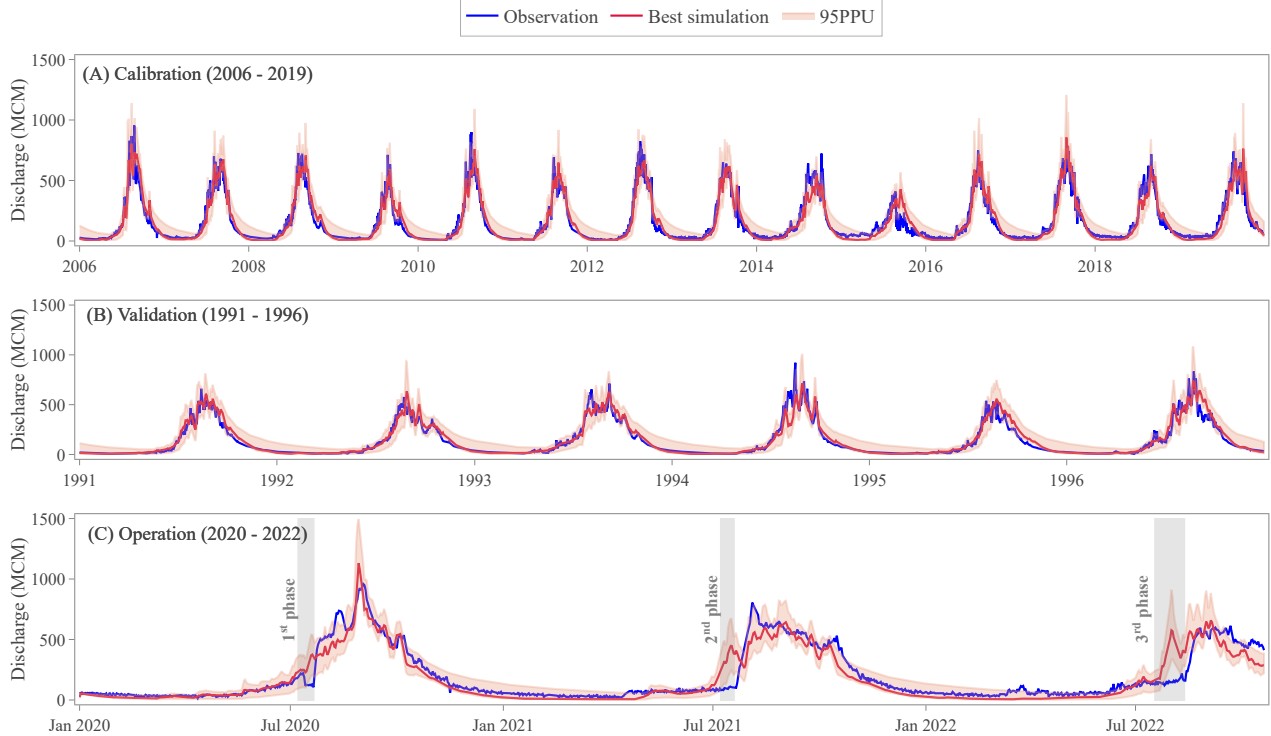

**Figure 4.** Temporal dynamics of daily simulated and observed discharge at the Upper Blue Nile basin outlet. Simulations forced with CHIRPS rainfall are shown during (A) the calibration period, (B) the validation period, and (C) the operation period. The best simulation was based on the parameter sets that achieved the highest NSE value during calibration. The 95% Prediction Uncertainty (95PPU) represents the 95[th] percentile of all very good simulations (NSE > 0.75) that was obtained by random generation of the parameter sets. The vertical grey shaded areas in the operation period roughly indicate the days in which the dam is filling. MCM stands for Million Cubic Meters.

On behalf of ten Nile Basin countries, the Nile Basin Initiative (NBI) has been commissioned to conduct studies in the
region. The Eastern Nile Technical Regional Office (ENTRO) of the NBI produced the E-A-S relationship of GERD reservoir and was published in Wheeler et al. (2016). When comparing the relationships derived from the DEM with the NBI, there is an agreement in the Elevation-Area relationship (see Fig. 5.A). However, the latter underestimates the storage corresponding to the water levels and surface areas (see Fig. 5.B and 5.C).

To validate the relationship, the intersected dotted grey lines in Fig. 5 demonstrate the reported characteristics of GERD by
the International Panel of Experts (IPOE), (IPoE, 2013). The most important for this study is the A-S relationship and especially for the areas less than 1000 km$^2$ in size, as GERD recently started to fill. For this range, the DEM-derived relationship intersects with the values reported by IPOE indicating better estimation than the NBI (see Fig. 5.B). Moreover, SRTM was commonly applied to obtain the E-A-S relationship of GERD (Kansara et al., 2021; Chen et al., 2021; Salama et al., 2022). For this reason, the following analysis will be based on the DEM-derived relationship.





**Table 4.** The statistic summary of HBV-light simulations.

| Product | NSE | RSR | PBIAS (%) | P-Factor (%) | R-Factor |
|---|---|---|---|---|---|
| **Calibration (2006 - 2019)** | | | | | |
| CHIRPS | 0.90 (Very good) | 0.32 (Very good) | −3.61 (Very good) | 77.84 | 0.73 |
| PERCIANN-CDR | 0.80 (Very good) | 0.45 (Very good) | −5.06 (Very good) | 54.98 | 0.40 |
| ARC2 | 0.79 (Very good) | 0.46 (Very good) | 1.53  (Very good) | 37.65 | 0.35 |
| **Validation (1991 - 1996)** | | | | | |
| CHIRPS | 0.91 (Very good) | 0.3  (Very good) | 4.64  (Very good) | 85.13 | 0.78 |
| PERSIANN-CDR | 0.74 (Good) | 0.52 (Good) | 32.20 (Unsatisfactory) | 49.18 | 0.50 |
| ARC2 | 0.81 (Very good) | 0.44 (Very good) | 22.88 (Satisfactory) | 58.02 | 0.39 |

The classification according to Moriasi et al. (2007) is as follows:

| Very good: | $0.75 < \text{NSE} \leq 1.0$; | $0 \leq \text{RSR} \leq 0.5$; | $\text{PBIAS} < \pm10$ |
|---|---|---|---|
| Good: | $0.65 < \text{NSE} \leq 0.75$; | $0.5 < \text{RSR} \leq 0.6$; | $\pm10 \leq \text{PBIAS} < \pm15$ |
| Satisfactory: | $0.50 < \text{NSE} \leq 0.65$; | $0.6 < \text{RSR} \leq 0.7$; | $\pm15 \leq \text{PBIAS} < \pm25$ |
| Unsatisfactory: | $\text{NSE} \leq 0.5$; | $\text{RSR} > 0.7$; | $\text{PBIAS} \geq \pm25$ |

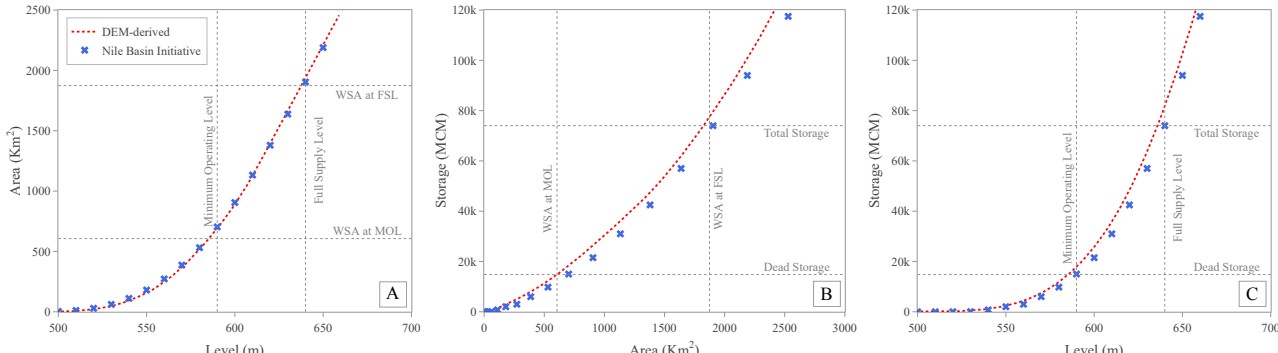

**Figure 5.** Elevation-Area-Storage (E-A-S) relationship of GERD reservoir obtained from SRTM digital elevation model in comparison with Nile Basin Initiative relationship. (A) Elevation-Area relationship, (B) Area-Storage relationship, and (C) Elevation-Storage relationship. The DEM-derived lines are the results of a fifth-degree polynomial fitting to the data points obtained at 1 m intervals of water levels.





### 4.3.2 Landsat-derived reservoir area and storage

The Water Surface Area (WSA) of the GERD reservoir was estimated using Landsat images using the algorithm modified by Vu et al. (2022) following the steps shown in Fig. 6.A. The algorithm uses the cloudless images (i.e., images with less than 20% clouds) to create an expanded mask and zone mask to correct the disturbed images. 133 Landsat images are available from January 2020 to September 2022, in which 70 images (53%) were classified as cloudless images. Fig. 6.B presents the WSA time series and revealed a significant increase in the WSA after improvement (green points) when compared with the WSA before improvement (light orange points). Additionally, the abrupt changes in the reservoir area are less observed after improvement, showing an important enhancement made by the algorithm. However, the WSA was underestimated in some images due to the high cloud coverage.

As can be seen in Fig. 6.B and 6.C, there are missing data during the wetting season (between June and September) of the three years which is due to the high cloud coverage. The first observed storage after the filling was on 19 August 2020 with a volume of 3.82 BCM and continued to decrease to 2.97 BCM on 11 June 2021. After the second phase, the storage jumped to 8.76 BCM (increased by 5.8 BCM) and gradually decrease till 2 March 2022 when a sudden decrease occurred. The water decreased from 8.06 BCM to reach 5.77 BCM by 3 April 2022. This decrease can be attributed to the announced 375 MW electricity generation that was started on 20 February 2022 according to Africa News (2022). In the third phase, the storage increased by 17 BCM reaching 23.1 BCM.

Furthermore, the increase in the WSA due to the filling was well captured by Landsat as illustrated in Fig. 7. The area was estimated to be 3.1% of the WSA at the Full Supply Level (FSL; 1874 km$^2$) before filling started and dramatically increased to reach 12%, 22%, and 44% during the first three filling phases, respectively. This indicates the considerable amount of water stored thus the importance of understanding the filling process for downstream water management.

The utilized algorithm was not always accurate in determining the threshold value of NDWI which in turn influenced the quantification of the area. Therefore, a visual inspection of the threshold was done to improve the estimation. Additionally, in fully cloud images the algorithm still provided additional water which is not true and needs to be checked manually.

### 4.3.3 Model-derived reservoir storage

The performance of storage volume quantification from HBV-light and the uncertainty due to model parameters and rainfall product selection are highlighted in Fig. 8. The very high modelled storage due to the use of PERSIANN-CDR and ARC2 resulted only in negative NSE values while CHIRPS was the only product that provided reliable reservoir storage as suggested by Fig. 8.A and 8.B. Around 629 runs of CHIRPS (out of 1756) have positive NSE values of which 407 simulations have NSE > 0.5. Additionally, the long filling periods suggested by ARC2 produced the highest RMSE values and range (between 44 and 78 BCM). Comparing the other two products, Fig. 8.C displays a higher and narrower interquartile range in the case of PERSIANN-CDR compared to CHRIPS. The performance of the best simulation of each rainfall product is exhibited in Table 5. Ultimately, the large difference between the rainfall products compared to the deviation caused by parameter uncertainty seen in Fig. 8, concludes that filling strategies are more influenced by input uncertainty than parameter uncertainty.



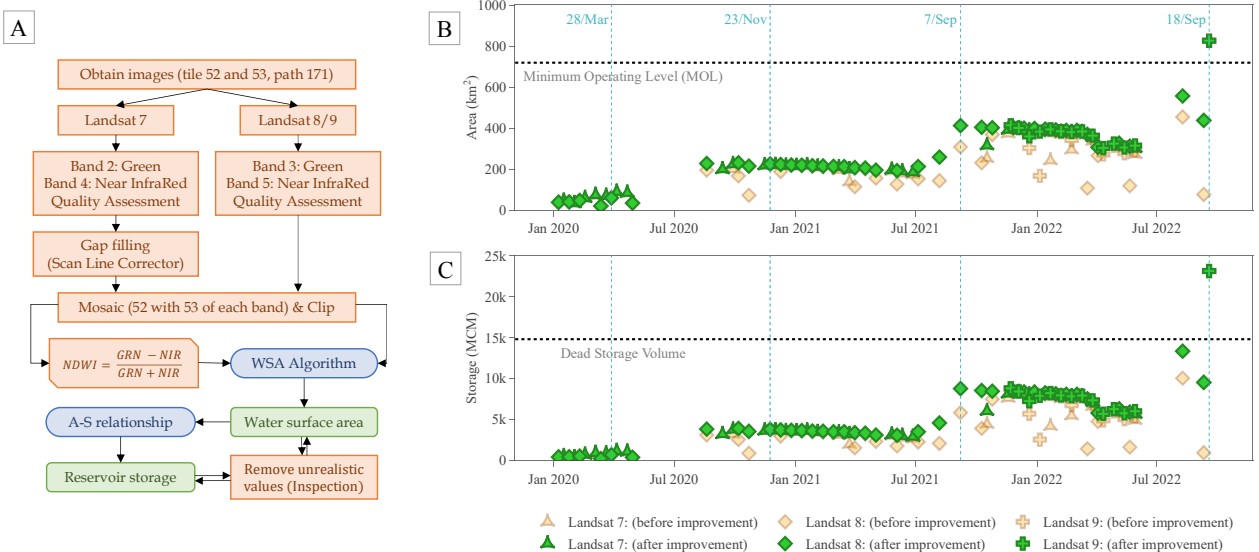

**Figure 6.** Procedure to derive GERD reservoir characteristics from Landsat. (A) The steps followed to derive the Water Surface Area (B) and Storage volume (C) of GERD reservoir from Landsat 7/8/9 Collection 2 Version 2 using the modified algorithm by Vu et al. (2022). The WSA maps of the dates specified in cyan dotted lines are illustrated in Fig. 7. Note that, not filling the gaps induced by the Scan Line failure of Landsat 7 produces NDWI = 0 in the gaps when using the algorithm. Therefore, in case of negative NDWI thresholds (which were the case here), Landsat 7 overestimates the areas and storages.

The modelled daily fraction of retained inflow, reservoir storage and area, and evaporated volume of water from HBV-light using CHIRPS are shown in Fig. 9 (for PERSIANN-CDR and ARC2 see Fig. C3). From the first row, it is clear that the

duration of filling increases with time suggesting around 13, 20, and 25 days for the first three phases, respectively. Looking at the storage retrieved from HBV-light shown in the second row, one can see the increase in the uncertainty through time which is due to the accumulation of discharge uncertainty with time. It is also important to note that the decrease witnessed by Landsat in 2022, due to the hydropower generation, was not captured by the model which does not include release/operation terms thus resulting in additional errors (i.e., lower RMSE can be achieved if this release was included). Additionally, the best simulation

suggests that the dead storage (14.79 BCM) was exceeded after the filling of 2022. The last two rows of Fig. 9 exhibit the daily area of the reservoir and the evaporation from the reservoir surface. Obviously, the evaporation increases with time due to the surface area increase which causes a slight gradual decrease in the storage.

## 4.4 Filling strategies and downstream discharge

### 4.4.1 GERD filling strategies

Fig. 10 exemplifies GERD filling strategies retrieved from CHIRPS since the other two rainfall products provided insufficient results. The figure shows a time series of monthly $\theta$ values and the percentage of reservoir filling. The $\theta$ values of May and





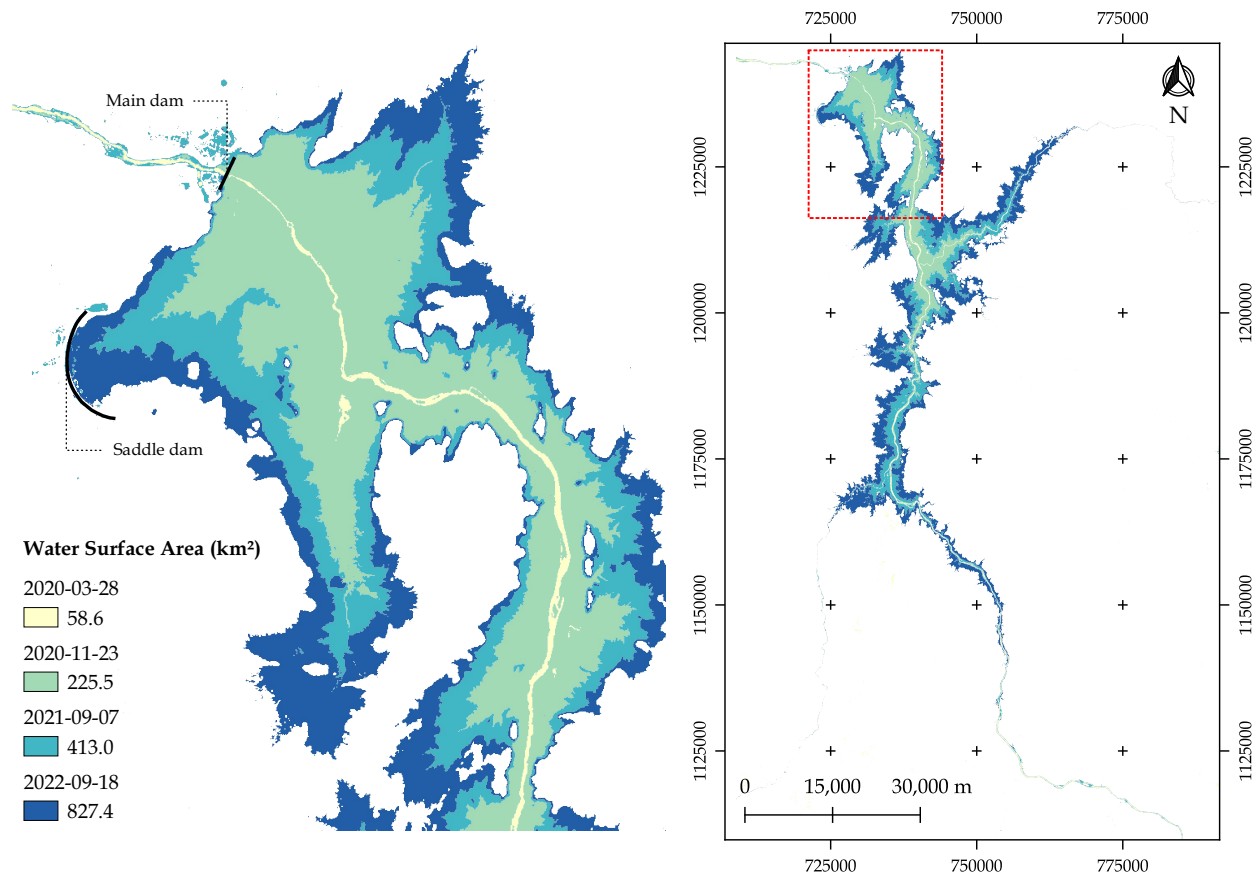

**Figure 7.** Landsat-derived water surface areas of Grand Ethiopian Renaissance Dam, using the modified algorithm by Vu et al. (2022), showing reservoir development through the first three filling phases from 2020 to 2022. The location of the zoomed area in the left figure is indicated by the rectangle (in red dotted line) in the right figure.

**Table 5.** The performance of HBV-Light best simulation in retrieving the storage volume of GERD reservoir taking Landsat-derived storage volume as reference.

| Satellite Rainfall Product | NSE | RMSE (BCM) |
| --- | --- | --- |
| CHIRPS | 0.77 | 1.70 |
| PERSIANN-CDR | −4.47 | 6.26 |
| ARC2 | −307.14 | 62.40 |




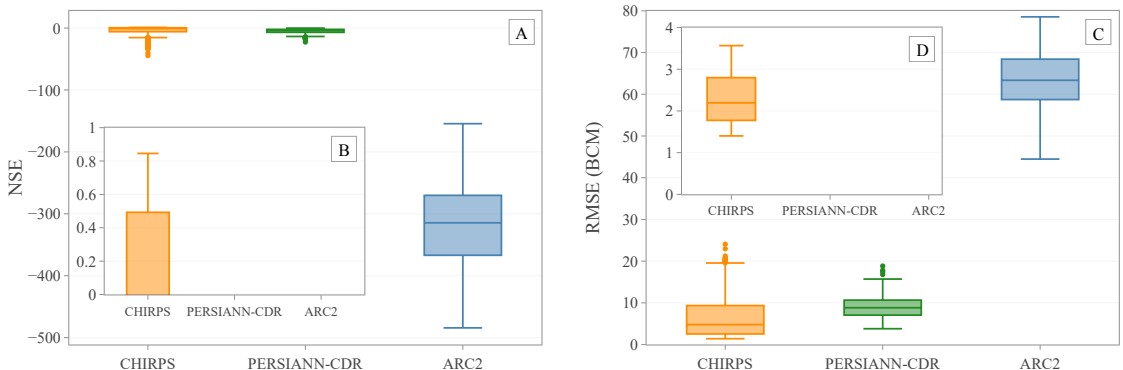

**Figure 8.** HBV-light performance in retrieving the storage volume of GERD from 2020 to 2022. (A, B) The Nash-Sutcliffe Efficiency (NSE) and (C, D) the Root Mean Square Error (RMSE) between the satellite observed storage (using Landsat) and the simulated GERD reservoir storage (using HBV-light). Panel (B) is a zoom-in of (A) showing NSE between 0 and 1 while (D) presents the RMSE of all simulations with NSE > 0. The x-axis is the three tested rainfall products for input uncertainty while the whiskers demonstrate HBV-light parameter uncertainty.

June 2020 are more than zero due to the low flows in these months and the uncertainty in the model (see also $\theta$ in Fig. 9). According to the red line, the filling of the three years started in July but continued till August in the year 2022. The percentage of the retained flow were found to be 14% in July 2020, 41% in July 2021, and 37% and 32% in July and August 2022,

respectively. Besides the best simulation, the parameter sets that reached NSE > 0.8 in retrieving the storage (Fig. 8.B) were able to well capture the filling dates. For other parameter sets, they suggest filling during other months (see green areas in Fig. 10). Moreover, in terms of annual volume, 5.2% and 7.4% of the total annual flow were retained in the first two years and between 12.9 - 13.7% in the following year. Retaining more than 10% of yearly flow in weeks is significant in terms of volume, and impacts downstream management. Due to parameter uncertainty, the minimum and maximum retained frictions

based on the 407 parameter sets (i.e., with NSE > 0.5) were written in Fig. 10. As far as the percentage of filling is concerned, HBV-light suggests that the filling reached 5.3% of the full storage capacity in the first year and considerably rose to reach 12.3% and 22.6% in 2021 and 2022, respectively.

### 4.4.2   Effect of GERD filling on the inflow to Sudan

With the filling strategies retrieved from the hydrological modelling, the observed daily discharge flowing to Sudan can be

better interpreted. Fig. 11 shows further analysis of the filling phases of GERD. At first, the discharge in the last three years showed unnatural behaviour when compared with the discharge before 2020 (see blue lines in Fig. 4). To investigate the role of GERD on the observed hydrograph, the difference in the inflow to Sudan after the construction of GERD and in case of no GERD was estimated and demonstrated in Fig. 11.A. The filling of GERD altered the hydrograph of the flowing water to Sudan. The monthly discharge to Sudan after 2020 showed a notable increase throughout the year except in the filling months

**Figure 9.** Temporal dynamics of inferred GERD filling strategies. The first three years of the Grand Ethiopian Renaissance Dam (GERD) filling strategies were retrieved from HBV-light lumped model using CHIRPS. The first row is the daily fraction of inflow volume retained by GERD. The second and third rows are the daily water storage and the corresponding water surface area of the GERD reservoir, respectively. The last row is the daily evaporated volume of water from the reservoir surface. The prediction uncertainty ranges indicate the 95, 60, 40, and 20 percentiles of the results obtained from the simulations classified as very good which were 1756.





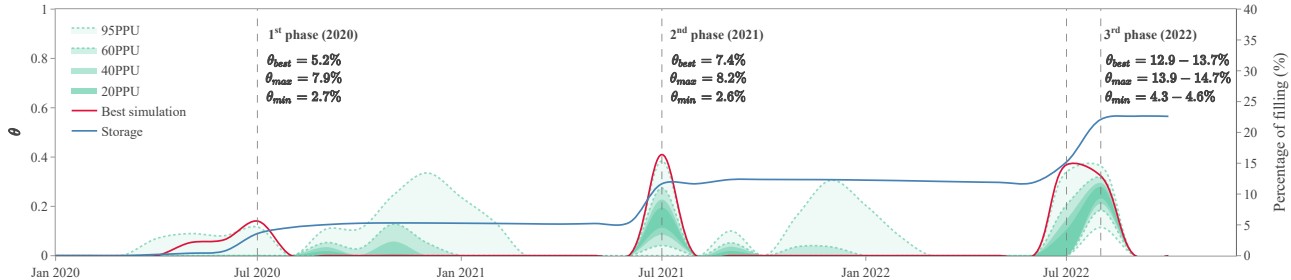

**Figure 10.** GERD filling strategies. $\theta$ expresses the fraction of the monthly inflow volume retained by the dam. The three values $\theta_{max}$, $\theta_{min}$, and $\theta_{best}$ (i.e., maximum, minimum, and best simulation) are the fraction of the annual inflow volume retained. The range in the values of the year 2022 is due to the unknown volume of November and December. The percentage of filling was calculated based on the best simulation and the GERD full storage capacity of 74 BCM. The uncertainty ranges (95 - 20 PPU) are based on the 407 runs with NSE > 0.5.

**Table 6.** Dates and volumes of the sudden changes in the downstream discharge after the completion of each GERD filling phase.

| Filling phase | Start | | End | | Average daily |
| --- | --- | --- | --- | --- | --- |
| | Date | Discharge (MCM) | Date | Discharge (MCM) | increase (MCM) |
| 1st phase (2020) | 21 July | 104.82 | 26 July | 489.18 | 76.87 |
| 2nd phase (2021) | 21 July | 100.40 | 4 August | 805.36 | 50.35 |
| 3rd phase (2022) | 13 August | 159.91 | 22 August | 522.90 | 40.33 |

(i.e., July 2020-2022, and August 2022) where the flow substantially decreased. This increase is partly due to heavy rains in 2020 and partly due to the operation of GERD. In terms of volume, the filling of the first two years did not influence the annual volume flowing to Sudan unlike in 2022. However, the decrease in the filling months influences the management of the Roseries and Sennar dams to store and divert enough water to the Gezira scheme and to meet other downstream demands.

Moreover, there are abnormal high flows caused by the highly intense rainfalls such as in August to October 2020 and May 375 2021 (see Fig. 11.B and 11.C). However, the considerable rise in the flowing water observed in March and April 2022 occurred as a result of the hydropower generation that started in February 2022 (see GERD release in Fig. 6.C). Paying attention to the storage change in Fig. 11.C, the relatively low outflows during filling days are followed by a sudden and steady increase in the outflow causing a linear hydrograph. Additionally, Table 6 illustrates the start and end dates of these changes and the magnitude of daily increase. The results revealed that the outflow increments are significant and decreased from around 77 MCM per day 380 in 2020 to 40 MCM per day in 2022.





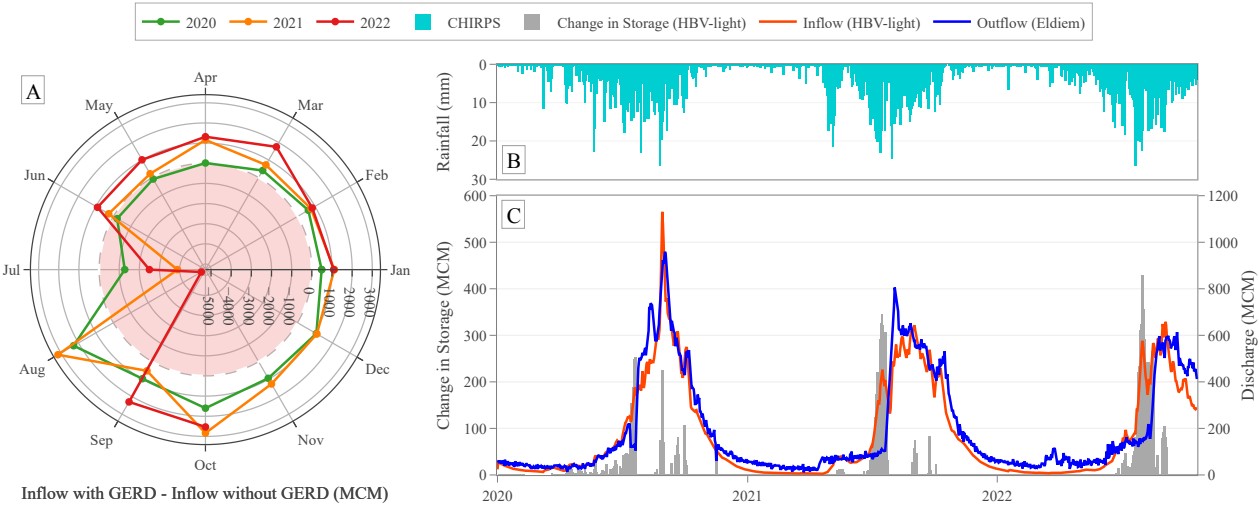

**Figure 11.** The impact of GERD filling on flow to Sudan: (A) the monthly impact of GERD on the incoming flow to Sudan: calculated as the difference between the inflow to Sudan with GERD and the inflow in case of no GERD, (B) The rainfall time series retrieved from CHIRPS, and (C) The daily change in storage of GERD reservoir and its implication on downstream daily discharge measured in Sudan. The change in storage is defined as the difference between two consecutive daily storage volumes.

## 5 Discussion

The unforeseeable construction progress of GERD complicates the predictability of the years required to completely fill the dam. Nevertheless, some trends can be expected based on the results of the current research. Retaining 20% of the average annual flow at the dam location (around 50 BCM) is equivalent to storing 10 BCM a year. Therefore, given the current stored 23.1 BCM of water and the retrieved friction of retained annual flow, GERD will need at least 5 more years (8 years in total) to complete filling under average flow conditions. However, a minimum of additional 3 years will be needed under wet conditions. During these years, it is important to reduce the risk of water diversions in Sudan by the adaptive management of reservoir operations (Wheeler et al., 2016). For the adaptive management of the Sudanese reservoirs, the current model can be extended to include the Lower Blue Nile basin and used to test operation scenarios for Roseires and Sennar dams. Other cooperative options for mitigating the effect of GERD on downstream flow, are coupling the operation of GERD with solar and wind power (Sterl et al., 2021) and applying water sharing policy during drought conditions (Wheeler et al., 2016; Yang and Block, 2021).

The current work might be of interest to studies on ungauged basins. One of the remaining challenges in ungauged basins is the need for more easily accessible data (Hrachowitz et al., 2013). The deficiencies of meteorological stations were replaced by a selection of a representative rainfall product based on the performance rankings adopted from Elagib and Mansell (2000). The ranking was found to give a first impression of the hydrological model performance to quantify the discharge as well as retrieve the filling strategies. Additionally, the calibration and validation of the hydrological model based on pre-operation years allowed for successful prediction of the inflow during post-operation years which was proved by the utilization of Earth





observation. Moreover, coupling the retrieved change in storage with the measured outflow of the dam allowed a further understanding of the downstream impact of the filling process (Vu et al., 2022).

Worldwide, there are 268 transboundary rivers covering 50% of the global land surface (Draper, 2007). Going a step forward, the novel methodology described in this paper can be adopted on a global scale, especially for data-scarce regions and transboundary rivers. However, discharge observations and the selection of representative precipitation products are crucial. The global implementation will raise the practical question: "How does the precipitation product will be selected? the best product globally? or a precipitation product per climate zone? or per reservoir?". Another important aspect is the reliability

of the current methodology. With the current error of around 1.7 BCM, the reliability increases with the increase of reservoir storage capacity. Thus, it is preferable to adopt the methodology for dams with large storage capacities. Nonetheless, further improvements in the methodology are needed to increase model efficiency and thus cover small reservoirs. The improvements are (but are not limited to):

–   Including more observations (e.g. soil moisture, groundwater measurements): with this additional data, independent val-
410        idation can be done which can help in eliminating many of the high-performed discharges due to their low performance on the other component (i.e., the additional used observation).

–   Validating both the precipitation and evapotranspiration when using remote sensing data: to minimize the uncertainty.

–   Utilizing actual evaporation measurements/validated product: to improve capturing the day-to-day variation in reservoir storage.

–   Including seepage term in the quantification of reservoir storage from reservoir water balance: new water levels are reached during filling, requiring more water initially than the obtained from the Elevation-Storage relationship to reach a certain level.

–   Using a more spatially representative hydrological model: daily variation in discharge can be better captured, but it requires extra information which may not be available.

**6   Conclusions**

The work done in this paper demonstrated the ability to utilize hydrological modelling with limited ground measurements to infer GERD filling strategies. The lack of information was compensated by the use of available Earth observation. From the evaluation of five selected rainfall products, CHIRPS was found to be the best-performing product in time (monthly) and space when compared with ten rain gauge stations, followed by PERSIANN-CDR, ARC2, GPCC, and ERA5, respectively.

Considering the top three products, HBV-light was calibrated and validated which resulted in high performance when the model was forced by CHIRPS, ARC2, and PERSIANN-CDR, respectively. During the operation period (2020 onwards), CHIRPS outperformed PERSIANN-CDR and ARC2 in capturing the filling properties (dates and magnitudes). The retrieved three years of GERD storage were compared with Landsat imagery, CHIRPS was the only product that produced positive NSE



values thus the lowest RMSE values. Subsequently, the selection of the rainfall product was found to have more influence
on inferring GERD filling strategies than parameter uncertainty. Additionally, the best simulation of CHIRPS (i.e., simulation
with the highest NSE value during calibration), was found to have an NSE of 0.77 and RMSE of 1.7 BCM. It inferred that
the friction of the monthly retained inflow was 14% for July 2020, 41% for July 2021, and 37% and 32% for July and August
2022, respectively. Moreover, around 5.2% and 7.4% of the annual inflow to Sudan were retained by the dam in 2020 and
2021, respectively. While after the third phase, in 2022, the reservoir retained between 12.9% and 13.7% of the annual inflow
reaching around 22.6% of the full storage capacity. Furthermore, it was revealed that the first three years started to impact the
flow downstream. During filling months, the inflow to Sudan significantly decreased below average (2002 - 2019), suggesting
the need for adapting reservoir management of the Roseires and Sennar dams to meet downstream demands.

*Data availability.* The publicly available data used in this study are the forcing data of the hydrological modelling (i.e., rainfall, temperature,
and potential evapotranspiration) as well as the satellite imagery. Landsat Collection 2 Level 2 was retrieved from https://earthexplorer.usgs.gov/.
The datasets of CHIRPS, PERSIANN-CDR and ERA5 were downloaded from the Climate Engine research app (Huntington et al., 2017). The
data of ARC2 is available in https://ftp.cpc.ncep.noaa.gov. Additionally, the GPCC dataset can be downloaded from https://opendata.dwd.de.
For the rain gauge data, it is not publicaly available but can be obtained from the Ethiopian National Meteorological Agency website
http://www.ethiomet.gov.et/ (navigate to "Data Service" and then select "Station Information" a request form will appear once a station has
been selected). The hydrological data within the Nile system is not publicly available due to governmental restrictions. However, the flow
data collected and maintained by the Nile Waters Directorate of the Ministry of Irrigation and Water Resources, Sudan, was made avail-
able for this study. Because we cannot share this data directly, we provide the successful HBV-light parameter sets and runs (NSE > 0.75)
as a best approximation of the observed flow at https://doi.org/10.4211/hs.ed4530307dda435e9d3dcdb74da86a30 in addition to the E-A-S
relationship derived from SRTM DEM (Ali, 2022)





## Appendix A: Additional data on performance metrics

**Table A1.** Statistical performance metrics applied in the evaluation of the satellite rainfall products and/or the simulated discharge and reservoir storage.

| Name | Symbol/Formula | Optimal value |
|------|----------------|---------------|
| Root Mean Square Error | $RMSE = \sqrt{\frac{1}{N}\sum_{i=1}^{N}(S_i - O_i)^2}$ | 0 |
| RMSE-observations Standard deviation Ratio | $RSR = \frac{RMSE}{\sigma_o}$ | 0 |
| Mean Bias Error | $MBE = \frac{1}{N}\sum_{i=1}^{N}(S_i - O_i)$ | 0 |
| Mean Absolute Error | $MAE = \frac{1}{N}\sum_{i=1}^{N}|S_i - O_i|$ | 0 |
| Percentage Bias | $PBIAS = \frac{\sum_{i=1}^{N}(S_i - O_i)}{\sum_{i=1}^{N} O_i} \times 100$ | 100 |
| Nash-Sutcliffe Efficiency | $NSE = 1 - \frac{\sum_{i=1}^{N}(S_i - O_i)^2}{\sum_{i=1}^{N}(O_i - \mu_o)^2}$ | 1 |
| Coefficient of Determination | $R^2 = \left(\frac{\sum_{i=1}^{N}(O_i - \mu_o)(S_i - \mu_s)}{\sqrt{\sum_{i=1}^{N}(O_i - \mu_o)^2(S_i - \mu_s)^2}}\right)^2$ | 0 |
| Slope | $Slope = \frac{(\sum_{i=1}^{N} O_i \cdot S_i) - \frac{(\sum_{i=1}^{N} O_i)(\sum_{i=1}^{N} S_i)}{N}}{(\sum_{i=1}^{N} O_i^2) - \frac{(\sum_{i=1}^{N} O_i)^2}{N}}$ | 1 |
| Intercept | $Intercept = \mu_s - Slope \times \mu_o$ | 0 |

$O$ = Observed value, $S$ = Gridded product or simulated value, $N$ = Number of samples, $\mu_o$ = Mean value of $O$, $\mu_s$ = Mean value of $S$, $\sigma_o$ = Standard deviation value of $O$, $\sigma_s$ = Standard deviation value of $S$. The coefficient of determination, intercept and slope are computed using linear fit.

**Figure A1.** Spatial distribution of the statistical performance metrics over the Upper Blue Nile basin based on the point-to-pixel approach at a monthly scale from 1984 to 2005. Six out of the ten stations are outside the UBN boundaries. However, all are included in the analysis to strengthen the evaluation of the rainfall products over the region. Basins boundaries and drainage network are obtained from the HydroSHEDS dataset (Lehner et al., 2008).





**Figure A2.** Statistical performance metrics of five remote sensing rainfall products evaluated at ten stations (coordinates in Table 3).





**Appendix B: Hydrological models applied in the Upper Blue Nile basin**

**Table B1.** Hydrological models applied over the Upper Blue Nile basin from 57 studies in the literature.

| No. | Model Name | No. of studies | References |
|---|---|---|---|
| 1 | SWAT | 23 | Betrie et al. (2009); Schmidt and Zemadim (2015); Wosenie (2015); Lemann et al. (2016); Roth and Lemann (2016); Chakilu and Moges (2017); Lemann et al. (2017); Polanco et al. (2017); Tegegne et al. (2017); Teklesadik et al. (2017); Woldesenbet et al. (2017); Lemann et al. (2018); Woldesenbet et al. (2018); Boru et al. (2019); Kessete et al. (2019); Nigussie et al. (2019); Sultana et al. (2019); Teshome et al. (2019); Adem et al. (2020); Sinshaw et al. (2020); Bizuneh et al. (2021); Getachew et al. (2021); Mengistu et al. (2021) |
| 2 | HBV | 13 | Ymeti (2007); Gragne et al. (2008); Wale et al. (2008); Uhlenbrook et al. (2010); Asnik (2015); Gebre et al. (2015); Meresa and Gatachew (2015); Teklesadik et al. (2017); Worqlul et al. (2017); Bihonegn et al. (2020); Bizuneh et al. (2021); Wubneh et al. (2022a, b) |
| 3 | HEC-HMS | 5 | Gebre (2015); Gebre and Ludwig (2015); Agegn (2016); Zelelew and Melesse (2018); Bihonegn et al. (2020) |
| 4 | PED | 5 | Enku et al. (2014); Zimale et al. (2016); Worqlul et al. (2017); Akale et al. (2019); Bihonegn et al. (2020) |
| 5 | CREST | 3 | Lakew et al. (2017); Lakew (2020); Lakew and Moges (2021) |
| 6 | SWIM | 2 | Aich et al. (2014); Teklesadik et al. (2017) |
| 7 | GR4J | 2 | Meresa and Gatachew (2015); Tegegne et al. (2017) |
| 8 | Thornthwaite and Mather (1955) | 2 | Collick et al. (2009); Legesse (2009) |
| 9 | FLexB | 1 | Wosenie (2015) |
| 10 | LAPSUS_D | 1 | Getahun (2016) |
| 11 | mHM | 1 | Teklesadik et al. (2017) |
| 12 | VIC | 1 | Teklesadik et al. (2017) |
| 13 | WaterGAP3 | 1 | Teklesadik et al. (2017) |
| 14 | CREST- SVAS | 1 | Lazin et al. (2020) |
| 15 | IHACRES | 1 | Tegegne et al. (2017) |
| 16 | JGrass-NewAge | 1 | Abera et al. (2017) |
| 17 | PCRaster | 1 | Tekleab et al. (2015) |
| 18 | Hydro-BEAM | 1 | Abd-El Moneim et al. (2017) |
| 19 | HBV-light-WEAP21 | 1 | Asitatikie and Gebeyehu (2021) |
| 20 | GBHM | 1 | Abdel-Aziz (2014) |
| 21 | MCDE | 1 | Nigussie et al. (2019) |
| 22 | SAC-SMA | 1 | Ymeti (2007) |
| 23 | RIBASIM | 1 | Ghorab et al. (2013) |
| 24 | HMETS | 1 | Meresa and Gatachew (2015) |
| 25 | TOPMODEL | 1 | Deginet (2008) |
| 26 | HBV-RIBASIM | 1 | Booij et al. (2011) |





## Appendix C: HBV-light simulaitons

**Table C1.** HBV-Light parameters, selected parameter ranges for Monte Carlo Simulations and optimized values of each selected satellite rainfall product.

| Parameter | Description | Unit | Range | | Optimize values | | |
|-----------|-------------|------|-------|-------|---------|--------------|------|
| | | | Min. | Max. | CHIRPS | PERSIANN-CDR | ARC2 |
| **Soil moisture routine** | | | | | | | |
| FC | Maximum soil moisture | mm | 200 | 1000 | 992.26 | 938.10 | 974.28 |
| LP | Soil moisture threshold for evaporation reduction | - | 0.5 | 0.7 | 0.528 | 0.510 | 0.520 |
| BETA | Shape coefficient | - | 1 | 4 | 1.78 | 1.43 | 1.06 |
| **Groundwater and response routine** | | | | | | | |
| PERC | Maximal flow from upper to lower GW-box | mm d$^{-1}$ | 1.4 | 2.8 | 1.71 | 1.85 | 1.48 |
| UZL | Threshold for K0-outflow | mm | 10.2 | 25.6 | 22.13 | 14.78 | 22.09 |
| K0 | Recession coefficient | d$^{-1}$ | 0.05 | 0.2 | 0.188 | 0.073 | 0.119 |
| K1 | Recession coefficient | d$^{-1}$ | 0.01 | 0.2 | 0.130 | 0.119 | 0.086 |
| K2 | Recession coefficient | d$^{-1}$ | 0.006 | 0.05 | 0.041 | 0.049 | 0.049 |
| **Routing routine** | | | | | | | |
| MAXBAS | Length of weighting function | d | 1.5 | 2.9 | 2.14 | 1.73 | 2.76 |

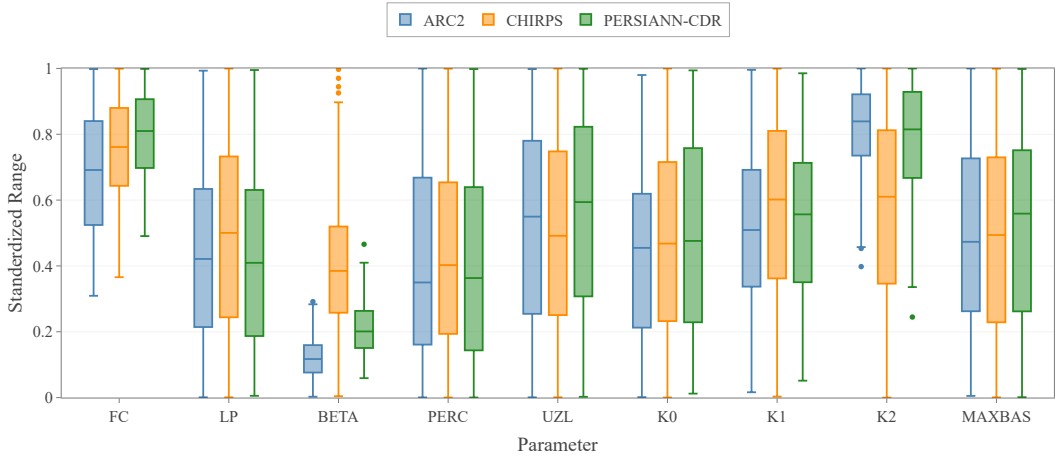

**Figure C1.** The standardized sensitive value ranges of HBV-light parameters for the simulations that classified as very good (i.e., NSE > 0.75) when using ARC2, CHIRPS, and PERSIANN-CDR as forcing data.





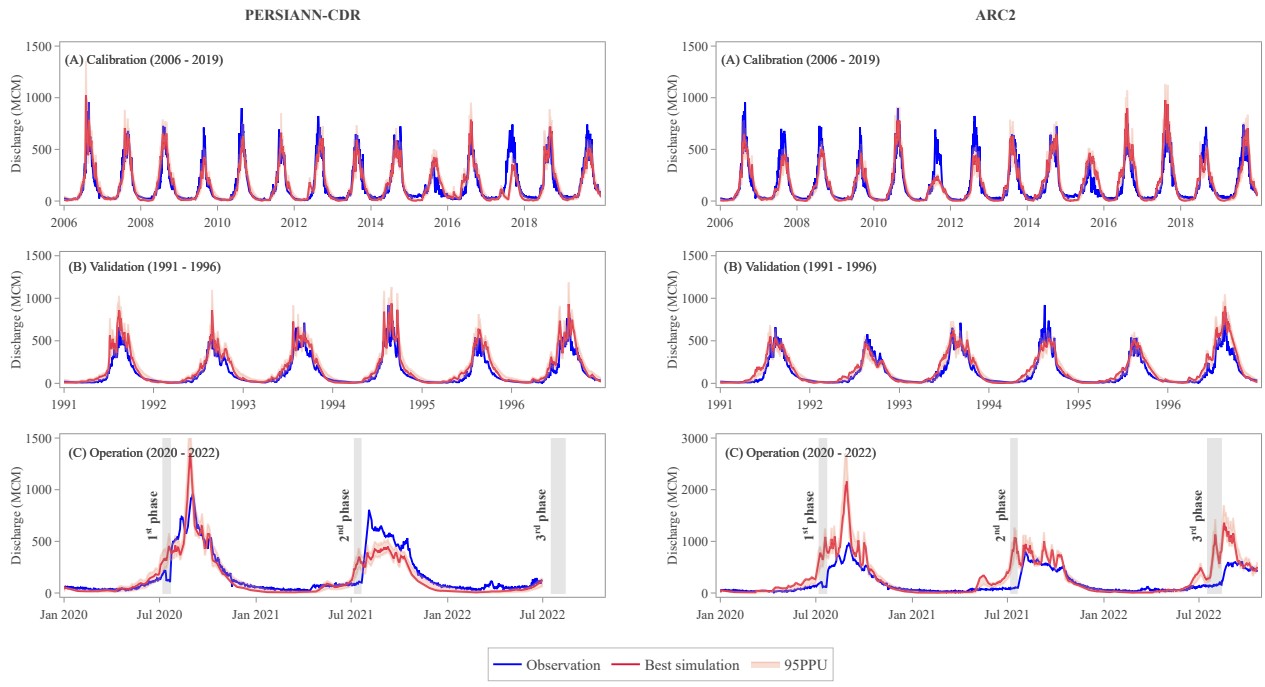

**Figure C2.** Temporal dynamics of daily simulated and observed discharge at the Upper Blue Nile basin outlet. Simulations forced with PERSIANN-CDR (left column) and ARC2 (right column) are shown during (A) Calibration period, (B) Validation period, and (C) Operation period. The best simulation was based on the parameter sets that achieved the highest NSE value during calibration. The 95% Prediction Uncertainty (95PPU) represents the $95^{th}$ percentile of all very good simulations (NSE > 0.75) that was obtained by random generation of the parameter sets. The vertical grey shaded areas in the operation period roughly indicate the days in which the dam is filling.



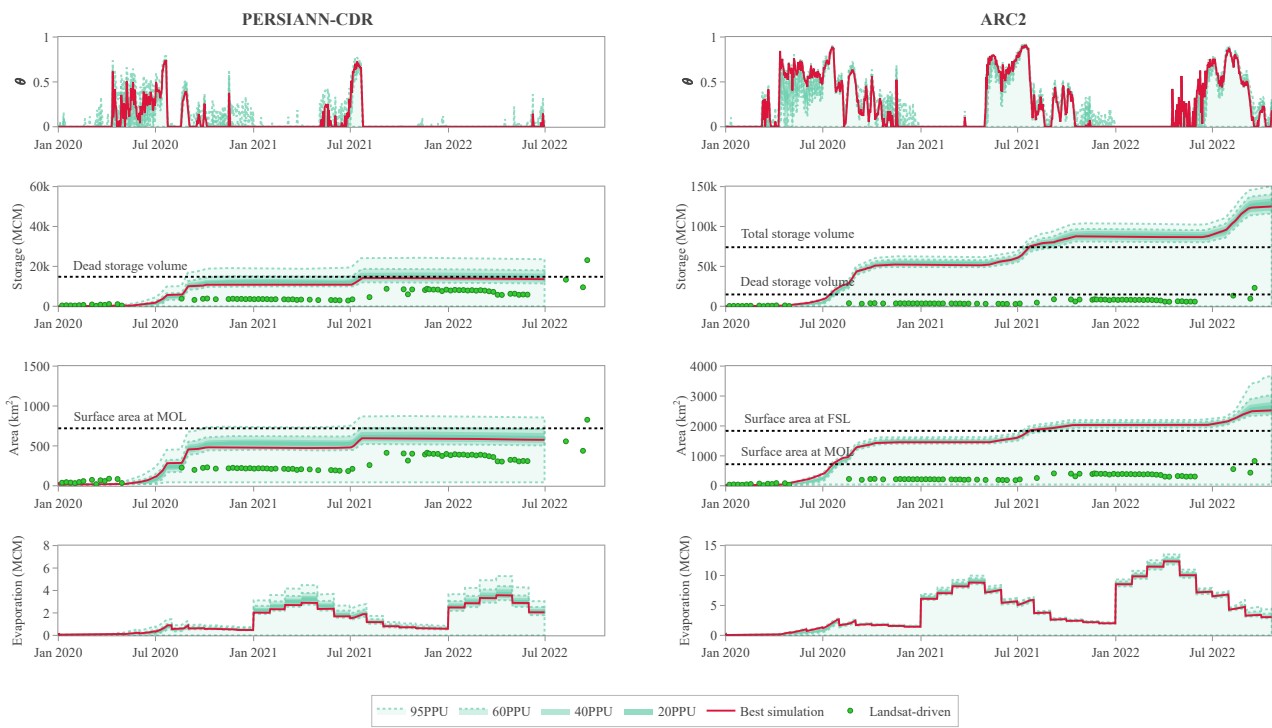

**Figure C3.** Temporal dynamics of inferred GERD filling strategies. The first three years of the Grand Ethiopian Renaissance Dam (GERD) filling strategies were retrieved from HBV-light lumped model using PERSIANN-CDR (left column) and ARC2 (right column). The first row is the daily fraction of inflow volume retained by GERD. The second and third rows are the daily water storage and the corresponding water surface area of the GERD reservoir, respectively. The last row is the daily evaporated volume of water from the reservoir surface. The prediction uncertainty ranges indicate the 95, 60, 40, and 20 percentiles of the results obtained from the simulations classified as very good which were 269 for PERSIANN-CDR and 244 for ARC2.





*Author contributions.* AMA was responsible for data curation, formal analysis, investigation, methodology development, acquiring resources, software, validation of data collection and analyses, visualization of results and writing of the original manuscript. AJT and LAM supervised the study which started as AMA's thesis towards a Master of Science degree. AJT acted as the primary supervisor and LAM as the secondary. All the authors contributed to the conceptualization, reviewing, and editing of the paper.

*Competing interests.* At least one of the (co-)authors is a member of the editorial board of Hydrology and Earth System Sciences, and the authors have also no other competing interests to declare.

*Acknowledgements.* This study was carried out as a master thesis by Awad M. Ali who is supported by the African Scholarship Programme (ASP) offered by the Wageningen University Fellowship Programme (WUFP). We also thank the Ministry of Irrigation and Water Resources of Sudan for providing the discharge data at Eldiem.



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
