# Peer review of "Inferring reservoir filling strategies under limited data availability using hydrological modelling and Earth observation: the case of the Grand Ethiopian Renaissance Dam (GERD)"

_Hydrology and Earth System Sciences, 2023_

## Author Comment (AC1)

**Response to referee comment: Anonymous Referee #1**

We greatly appreciate Anonymous Referee #1 for the time and effort put to review our manuscript and for the constructive comments. We believe it will help improving and better clarifying our work. Please see below our point-to-point response.

**Comment:** The paper contributes a reconstruction of the filling strategy of the Grand Ethiopian Renaissance Dam (GERD) in Ethiopia based on a combination of hydrological modeling and satellite data processing. The approach also explores the role of precipitation data uncertainty by considering five alternative rainfall products as input to the conceptual hydrological model (HBV). The study is very timely and interesting for HESS readership. However, despite the modeling framework looks solid and well designed, its implications and value for the ongoing water management dispute in the Nile River Basin should be better elaborated (see comments below) before accepting the paper for publication.

1) Since the abstract and in most of the introduction, the authors emphasize the value of inferring the GERD filling strategy to support a better management of the Nile River despite the long-lasting international tensions between Ethiopia and downstream countries (see lines 19-22; 57-58 74-75). I do second this point but would argue that the paper falls shy of these contributions to water management. The proposed approach uses a conceptual hydrological model calibrated at the Eldiem station (close to the Ethiopia-Sudan border) before the GERD construction to estimate the volume of water stored during the filling period as the difference between the simulated discharge in natural conditions minus the observed one. The resulting storage trajectory is validated against a trajectory reconstructed from Landsat images according to the method proposed by Vu et al. (2022). As said before, this modeling approach sounds solid and well-designed, except for a couple of minor points reported below.

**Response:** We believe that our manuscript delivers on its promise by offering valuable insights to improve the management of the two downstream dams in Sudan, particularly in light of the current dispute and the lack of shared GERD filling plans. No other study has presented quantitative information about the impact of GERD on downstream discharge, which is crucial for water managers in Sudan to optimize the operation of the dams for food security and hydropower generation. With the information provided in our manuscript, it is possible, for example, to reschedule the filling of the Roseires dam to align with agricultural requirements in the coming years (discussed in line 388-389). However, while our aim is to provide this necessary information, we do not intend to apply it to optimize current management practices since this would require significant additional work.

Page 3 line 75: "Blue Nile River" is replaced by "Lower Blue Nile River".

Page 3 line 75: after "…. in Sudan." we have added "Nonetheless, the work does not attempt to use the findings to optimize current management practices since this would go beyond the goal of this study."

**Comment:** However, the discussion in Section 4.4 about the value of these findings for the ongoing water management dispute in the Nile River Basin is relatively simplistic. Here, the authors only comment about the reconstructed filling strategy (Fig. 10) and streamflow entering in Sudan (Fig. 11), raising the following doubt: is the proposed approach really necessary for informing water

management? On the one hand, the reconstructed trajectory in Fig. 10 could be obtained with the approach by Vu et al (2022) only using satellite images; on the other hand, the flow entering in Sudan is directly measured at Eldiem station, regardless of the models developed for the upstream part.

**Response:** While the approach presented by Vu et al. (2022) is useful, we acknowledge that it has certain limitations that we have addressed in our manuscript. In the case of GERD, we believe that having a fine time scale (daily or less) is crucial, especially given the proximity of the Roseires dam to GERD.

Page 22 line 399: we have added "Beside the approach discussed here, the downstream analysis can also be based on satellite images using, for example, the approach propose by Vu et al. (2022). However, relying on the latter approach for real-time operation presents certain challenges. Firstly, given the current availability of free satellite data (such as Sentinel and Landsat), it is not feasible to achieve daily time steps, unlike in the case of relying on hydrological modelling. Secondly, waiting for a few days to acquire satellite data can be problematic, particularly during flood events, such as those experienced in Sudan in 2020. It is important to note that our proposed approach relies on outflow observations, which may not always be available. As such, both approaches have their respective applications and limitations."

**Comment:** To satisfy the (high) expectations generated in the abstract-introduction, I believe the authors should try to expand this part of the manuscript in order to better show the potential value of their model. For example, can you use your results to infer a rule that could be used to simulate the rest of the filling period? can you quantify the value of the information produced by your model for supporting the pro-active operations of Roseires and Sennar dams (as mentioned at line 67)? how should these two dams be operated to adapt/coordinate with the upstream filling policy?

Replying to this type of questions is in my opinion necessary to make the paper's findings valuable on the policy side. If authors believe this is going beyond the scope of their work, I would suggest revising the narrative of the abstract and intro in order to downplay these aspects and better characterize their contribution.

**Response:** Regarding the statement in line 67 of our manuscript, our intent was to provide justification for the proposed methodology. However, we acknowledge that the first two questions raised by the referee are integral to the objectives of our study. While we believe that our methodology could provide valuable insights on these questions, we also recognize that their answers are uncertain given the unfinished construction of GERD, as we have discussed in line 382. In practice, if the construction of GERD dam was completed, these questions could be addressed using our methodology, which involves utilizing stochastic hydrological simulations based on historical discharge data. Additionally, we declare that a more effective management of dams could be achieved by understanding past filling strategies, particularly given the uncertainty surrounding future filling strategies based on the information currently available. As for the third question, it is addressed in our response to the first comment.

**Comment:** 2) The reconstruction of the filling strategy is built on the hydrologic simulation of the HBV-light model. This conceptual model was calibrated during the 2006-2019 period and validated over the period 1991-1996. How reliable is this strategy given the evident global warming/climate change trends? Did the authors check the presence of trends in precipitation and temperature data?

**Response:** In our manuscript, we explicitly acknowledged in line 165 that the approach we took was necessitated by a notable deficiency in the available data. Given this constraint, we made a deliberate decision to partition the periods in this way. We appreciate your suggestion to consider the impact of global warming on our findings, and we would like to address your concerns regarding this matter.

We are confident that the findings based on the current strategy are reliable for two reasons. First, the change in climate between calibration and validation is likely much larger than between calibration and the present, since these periods are much closer in time. Second, as noted by Melsen et al. (HESS, 2018), parameter sets are not typically the dominating source of uncertainty when considering long-term changes simulated with hydrological models such as HBV, which was also employed in this study.

**Comment:** Since the calibration relies on 10,000 random parameter sets that returned 1756 simulations with NSE>0.75 (lines 242-244), I suspect the "best" parameterization adopted might not necessarily be so valid when applied to the 2020-2022 time period.

**Response:** As stated in lines 250-255, the value of NSE is greatly influenced by the discharge seasonality of the basin. As such, we only obtained 407 simulations with an NSE greater than 0.5 in retrieving the storage, as indicated in lines 332-333. We believe that these 407 simulations are more valid than the 1756 simulations. Our results indicate that the "best" parameterization is reasonably valid since it achieved the highest performance during the calibration period and obtained a high NSE value in retrieving the storage. However, we acknowledge that other parameterization options may also be valid, but we have attempted to make our methodology as objective as possible.

**Comment:** 3) The authors are validating the reconstruction based on the HBV-light model using the approach by Vu et al. (2022). However, they notice only 53% of the Landsat images are cloudless, with several missing data during the wet season, which is also the most critical in terms of filling. Why did they not consider also using radar altimetry data to complement Landsat images?

**Response:** The Global Reservoirs and Lakes Monitor (G-REALM; Birkett et al., 2011) has actually been part of our methodology initially, but as it turned out to be too uncertain it was excluded. As depicted in Fig. 1(a) below, the measurement location provided in G-realm (Jason and Sentinel) was about 80km upstream of the dam body. The storage volume derived from G-realm had an unreasonable rapid decline after peak and was high compared to Landsat data (see Fig. 1(b) below). As a result, we validated only using Landsat data. Additionally, radar altimetry data will only add a few points to the time series, which is not enough to be useful. We hope the reviewer agrees with our motivation not to consider altimetry. However we will make sure to discuss the potential of altimetry in the discussion section.

[Figure]

**Fig. 1:** Panel (a): dam body and G-realm measurement location (Birkett et al., 2011), Panel (b): GERD storage volume obtained using G-realm (blue) and Landsat (before improvement: orange, after improvement: green).

**References:**

Birkett, C., Reynolds, C., Beckley, B., & Doorn, B. (2011). From research to operations: The USDA global reservoir and lake monitor. Coastal altimetry, 19-50.

Melsen, L. A., Addor, N., Mizukami, N., Newman, A. J., Torfs, P. J., Clark, M. P., ... & Teuling, A. J. (2018). Mapping (dis) agreement in hydrologic projections. Hydrology and Earth System Sciences, 22(3), 1775-1791.

Vu, D. T., Dang, T. D., Galelli, S., & Hossain, F. (2022). Satellite observations reveal 13 years of reservoir filling strategies, operating rules, and hydrological alterations in the Upper Mekong River basin. Hydrology and Earth System Sciences, 26(9), 2345-2364.

---

## Author Comment (AC2)

Hess-2023-19

**Response to referee comment: Anonymous Referee #2**

We thank Anonymous Referee #2 for the detailed review and his/her support for our work. We trust considering his/her feedback will be valuable for our work. Here we provide point-to-point responses to your comments.

**Comment:** Summary: This study uses a lumped hydrological model, along with (not "coupled with" as noted in the abstract) remote sensing data to examine the filling strategies of the Grand Ethiopian Renaissance Dam (GERD). The model used is the HBC-light model, which is used to simulate the inflow into the reservoir, evaporation etc. Overall, it is an interesting study that presents substantial results on inflow, outflow, and filling strategies for GERD. The paper is generally well written and there is a lot to like in the paper, thus I am generally supportive of the work and believe that it could eventually be published in HESS; however, substantial revisions are necessary before the paper can be accepted. I provide my detailed comments below.

\*\* Abstract, Line 6, "coupled": I don't think the model is coupled with remote sensing data. RS data is used in conjunction with the model. Please revise this statement.

**Response:** "coupled" will be changed to "in conjunction with"

**Comment:** \*\* Abstract and conclusion: As I noted above, there is a lot going on in this study; however, I am not convinced that the study, at least as it stands in the current form, presents sufficient novel scientific insights. It surely presents substantial information that could be used to manage reservoirs in the study region, but I ask: what is the scientific merit? I suggest that authors revise the introduction to address this issue, and perhaps some changes in the results and conclusion sections should be made as well.

\*\* End of introduction (Lines 70-79): this is not very convincing. Again, what is the major scientific contribution of this work? Please clearly specify scientific questions and objectives. The authors attempt to justify the study (toward the end of the paper) noting that the approach/framework could be generally applicable to other (data sparse) regions; I am not sure how valid this claim is given substantial uncertainties in the ability to simulate the flows by the model and the inherent limitations in remote sensing data.

**Response:** The introduction and conclusion will be revised to better clarify the novelty and address the issue raised by the reviewer.

**Comment:** \*\* The simulated inflow is somewhat questionable as it is not validated with any observed data. Given many sources of uncertainty, how do the authors ensure that the simulated inflow is reasonable?

**Response:** The model was validated with observed data at Eldiem station. This station is close to the dam location, hence we assume that the discharge in both locations (before dam construction) are equal.

Calibration and validation were done prior to dam construction to ensure adequate representation of catchment hydrology. The calibrated model was then used to estimate dam inflow, which was proved to be reliable based on good agreement with dam storage data retrieved from Landsat.

**Comment:** ** Figure 4 (related to the above comment), "Best simulation": I assume "observation" here is the outflow and "Best simulation" is the inflow. Please clarify by changing the legends.

**Response:** P13 Figure 4: "Observation" will be changed to "Outflow (observation)" and "Best Simulation" to "Inflow (best simulation)"

**Comment:** ** Figure 4: Is the unit "MCM"? Discharge should have a unit with per unit time, not just volume! Same applies to the right axis of Figure 11.

**Response:**

P13 Figure 4: the unit "MCM" will be changed to "MCM d$^{-1}$"

P21 Figure 11: the unit "MCM" will be changed to "MCM d$^{-1}$"

**Comment:** ** Figure 7 and other relevant section: Given limitations in Landsat imageries, could the authors use other remote sensing products such as those from Sentinel?

**Response:** We will try to retrieve water surface extent using Sentinel to provide some data points during wet season as suggested by the referee.

**Comment:** ** There are high uncertainties/errors in many of the results presented, which are not discussed in the present manuscript. The uncertainties arising from precipitation data are discussed, but there are many other sources of uncertainties including the use of PET data from ERA5. Is the PET data reliable? How do any uncertainties in the PET data affect the results. I suggest that the authors present a dedicated (concise) section on various sources of uncertainties, the implication on the results, and potentially the conclusion drawn.

**Response:** The discussion will be extended to include all points mentioned by the referee.

**Comment:** ** L403: The question here suffers from critical grammatical issues. Please carefully rewrite it.

**Response:** P22 L403 : the questions will be corrected to "How is the precipitation product selected?"

**Comment:** Minor comments:

\*\* There are excessive abbreviations used in this paper. I suggest removing those that are not necessary.

**Response:** we will revise the abbreviations and unnecessary ones will be removed.

**Comment:** \*\* I found that figure number is not done in an increasing order in the text.

**Response:** This is mainly in L180 and will be revised.

Also, the arrangement/placement of figures in the text is not good; many figures are 2-3 pages away from where they are referenced in the text, which makes it hard to read.

**Response:** We agree; it will be corrected in the revised version.

**Comment:** \*\* L85: UBN was already spelled out.

**Response:** P3 L85: "the Upper Blue Nile (UBN) basin" will be replaced by "the UBN basin""

**Comment:** \*\* L118: What does "satellite imagery" mean here? Please specify.

**Response:** P5 L118: after "… satellite imagery" we will add "(Landsat 7,8, and 9)"

**Comment:** \*\* L210: "is" à "are"?

**Response:** P9 L210: "is" will be corrected to "are"

**Comment:** \*\* L248: "shows" = "show"

**Response:** P11 L248: "shows" will be corrected to "show"

---

## Author Response (AR1)

Hess-2023-19

**Inferring reservoir filling strategies under limited data availability using hydrological modelling and Earth observation: the case of the Grand Ethiopian Renaissance Dam (GERD)**

Dear Editor,

We would like to thank you and the two anonymous reviewers for their favourable comments and constructive feedback. We appreciate the time and effort they invested in providing us with valuable insights. We are pleased to hear that both reviewers found our work interesting and well-structured. However, we acknowledge the suggestions made by reviewer #1 and reviewer #2, and have made significant efforts to address their concerns in this revised version.

Reviewer #1 suggests better elaborating on the implications of the results on the Nile water management. While reviewer #2 suggested adding another satellite imagery to address Landsat limitations, as well as better clarify the novelty and the scientific contribution. Therefore, here we provide a new version of the manuscript trying to address the points raised by the reviewers. Mainly, the manuscript included additional data and analysis regarding Sentinel imageries, thus updating relevant figures. Additionally, we extended the discussion to include the implication, uncertainty and novelty of this study.

We hope that these revisions address the concerns raised by the reviewers and improve the quality of our manuscript.

Best regards,

Awad M. Ali (on behalf of the authors)

**Response to referee comment: Anonymous Referee #1**

\* Please note that page and line numbers in the response are based on the updated manuscript!

**Comment:** The paper contributes a reconstruction of the filling strategy of the Grand Ethiopian Renaissance Dam (GERD) in Ethiopia based on a combination of hydrological modeling and satellite data processing. The approach also explores the role of precipitation data uncertainty by considering five alternative rainfall products as input to the conceptual hydrological model (HBV). The study is very timely and interesting for HESS readership. However, despite the modeling framework looks solid and well designed, its implications and value for the ongoing water management dispute in the Nile River Basin should be better elaborated (see comments below) before accepting the paper for publication.

1) Since the abstract and in most of the introduction, the authors emphasize the value of inferring the GERD filling strategy to support a better management of the Nile River despite the long-lasting international tensions between Ethiopia and downstream countries (see lines 19-22; 57-58 74-75). I do second this point but would argue that the paper falls shy of these contributions to water management. The proposed approach uses a conceptual hydrological model calibrated at the Eldiem station (close to the Ethiopia-Sudan border) before the GERD construction to estimate the volume of water stored during the filling period as the difference between the simulated discharge in natural conditions minus the observed one. The resulting storage trajectory is validated against a trajectory reconstructed from Landsat images according to the method proposed by Vu et al. (2022). As said before, this modeling approach sounds solid and well-designed, except for a couple of minor points reported below.

**Response:** We believe that our manuscript delivers on its promise by offering valuable insights to improve the management of the two downstream dams in Sudan, particularly in light of the current dispute and the lack of shared GERD filling plans. No other study has presented quantitative information about the impact of GERD on downstream discharge, which is crucial for water managers in Sudan to optimize the operation of the dams for food security and hydropower generation. With the information provided in our manuscript, it is possible, for example, to reschedule the filling of the Roseires dam to align with agricultural requirements in the coming years. However, while we aim to provide this necessary information, we do not intend to apply it to optimize current management practices since this would require significant additional work.

P4 L77: The results intend to support the management of the Sudanese dams so to make it more clear we add "Lower" before "Blue Nile River".

P4 L77: To lower the readers expectation of our study, after "…. in Sudan." we added "Nonetheless, the work does not attempt to use the findings to optimize current management practices since this would go beyond the goal of this study."

**Comment:** However, the discussion in Section 4.4 about the value of these findings for the ongoing water management dispute in the Nile River Basin is relatively simplistic. Here, the authors only comment about the reconstructed filling strategy (Fig. 10) and streamflow entering in Sudan (Fig. 11), raising the following doubt: is the proposed approach really necessary for informing water management? On the one hand, the reconstructed trajectory in Fig. 10 could be obtained with the approach by Vu et al (2022) only using satellite images; on the other hand, the flow entering in Sudan is directly measured at Eldiem station, regardless of the models developed for the upstream part.

**Response:** While the approach presented by Vu et al. (2022) is useful, we acknowledge that it has certain limitations that we have addressed in our manuscript. In the case of GERD, we believe that having a fine time scale (daily or less) is crucial, especially given the proximity of the Roseires dam to GERD. Moreover, our approach also allowed to provide crucial information about the impact of the dam by testing the difference between with and without GERD scenarios (See Fig. 11.A). The latter analysis is not possible when relying only on satellite imagery.

P23 L442: we added "It is worth mentioning the main advantages of the proposed approach. Besides the approach discussed here, the downstream analysis can also be based on satellite images using, for example, the approach as proposed by Vu et al. (2022). However, relying on the latter approach for real-time operation presents certain challenges. Firstly, given the current availability of free satellite data (such as Sentinel and Landsat), it is not feasible to achieve daily time steps, unlike in the case of relying on hydrological modelling. Secondly, waiting for a few days to acquire satellite data can be problematic, particularly during flood events, such as those experienced in Sudan in 2020. Moreover, hydrological modelling allows to investigate future scenarios under different management strategies. It is important to note that our proposed approach relies on outflow observations, which may not always be available/accessible. As such, both approaches have their respective advantages and limitations."

**Comment:** To satisfy the (high) expectations generated in the abstract-introduction, I believe the authors should try to expand this part of the manuscript in order to better show the potential value of their model. For example, can you use your results to infer a rule that could be used to simulate the rest of the filling period? can you quantify the value of the information produced by your model for supporting the pro-active operations of Roseires and Sennar dams (as mentioned at line 67)? how should these two dams be operated to adapt/coordinate with the upstream filling policy?

Replying to this type of questions is in my opinion necessary to make the paper's findings valuable on the policy side. If authors believe this is going beyond the scope of their work, I would suggest revising the narrative of the abstract and intro in order to downplay these aspects and better characterize their contribution.

**Response:** We acknowledge that the first two questions raised by the referee are integral to the objectives of our study. While we believe that our methodology could provide valuable insights on these questions, we also recognize that their answers are uncertain given the unfinished construction of GERD, as we have discussed in L423. In practice, if the construction of GERD dam was completed, these questions could be addressed using our methodology, which involves utilizing stochastic hydrological simulations based on historical discharge data. Additionally, we declare that a more effective management of dams could be achieved by understanding past filling strategies, particularly given the uncertainty surrounding future filling strategies based on the information currently available. As for the third question, it is addressed in our response to the first comment.

Therefore, we added a paragraph that demonstrates the implications and outlook of our findings on the management of the dams in the Lower Blue Nile river.

P22 L432: "The operation of downstream dams (i.e., Roseires and Sennar) depends on the management of the GERD dam. Thus, the outcomes of this approach provide information that can support downstream dam operations. The findings of this study indicate that lower discharge levels were observed during July and August, while higher discharge rates were observed during other periods. It suggests that additional water should be stored during GERD-non-filling periods to achieve the targeted monthly elevation of the Roseires reservoir. A substantial increase in discharge is noted during March-June and October-December. We propose that effective management of Roseires and Sennar should

involve storing additional water during these months to compensate for shortages during filling periods. Additionally, for adaptive management of Sudanese reservoirs, the existing model should be expanded to include the Lower Blue Nile basin and used to evaluate alternative operation scenarios for Roseires and Sennar dams. This approach will enable a comprehensive understanding of the implications of various management decisions and facilitate the development of optimized reservoir management plans."

**Comment:** 2) The reconstruction of the filling strategy is built on the hydrologic simulation of the HBV-light model. This conceptual model was calibrated during the 2006-2019 period and validated over the period 1991-1996. How reliable is this strategy given the evident global warming/climate change trends? Did the authors check the presence of trends in precipitation and temperature data?

**Response:** In our manuscript, we explicitly acknowledged in L172 that the approach we took was necessitated by a notable deficiency in the available data. Given this constraint, we made a deliberate decision to partition the periods in this way. We appreciate your suggestions to consider the impact of global warming on our findings, and we would like to address your concerns regarding this matter.

P8 L173: We added "Although the climate might show non-stationarity, we still consider the optimized parameters robust enough to draw conclusions for two reasons. First, the calibration and validation periods extend over a long duration. Second, as noted by Melsen et al. (2018), parameter sets are not typically the dominating source of uncertainty when considering long-term changes simulated with hydrological models such as HBV, which was also employed in this study."

**Comment:** Since the calibration relies on 10,000 random parameter sets that returned 1756 simulations with NSE>0.75 (lines 242-244), I suspect the "best" parameterization adopted might not necessarily be so valid when applied to the 2020-2022 time period.

**Response:** As stated in L269-271, the value of NSE is greatly influenced by the discharge seasonality of the basin. As such, we only obtained 407 simulations with an NSE greater than 0.5 in retrieving the storage, as indicated in L358-359. We believe that these 407 simulations are more valid than the 1756 simulations. Our results indicate that the "best" parameterization is reasonably valid since it achieved the highest performance during the calibration period and obtained a high NSE value in retrieving the storage. However, we acknowledge that other parameterization options may also be valid, but we have attempted to make our methodology as objective as possible.

P12 L262: We added "In further analysis, we rely on the parameter set that resulted in the best model performance (best simulation). However, we acknowledge that other parameter sets with a high performance may be equally valid."

**Comment:** 3) The authors are validating the reconstruction based on the HBV-light model using the approach by Vu et al. (2022). However, they notice only 53% of the Landsat images are cloudless, with several missing data during the wet season, which is also the most critical in terms of filling. Why did they not consider also using radar altimetry data to complement Landsat images?

**Response:** The Global Reservoirs and Lakes Monitor (G-REALM; Birkett et al., 2011) has actually been part of our methodology initially, but as it turned out to be too uncertain it was excluded. As depicted in Fig. 1(a) below, the measurement location provided in G-realm (Jason and Sentinel) was about 80km upstream of the dam body. The storage volume derived from G-realm had an unreasonable

rapid decline after peak and was high compared to Landsat data (see Fig. 1(b) below). As a result, we validated only using Landsat data. Additionally, radar altimetry data will only add a few points to the time series, which is not enough to be useful. We hope the reviewer agrees with our choice to not include altimetry data after careful consideration. We mention the potential use of altimetry in section 4.3.2. Alternatively, as this point was also raised by referee #2, we added Sentinel-2 data to compensate for the limitations of Landsat (see details in response to referee #2 comments).

P16 L349: We added "It is important to point out that open-access radar altimetry datasets provide valuable information for this validation step. Therefore, for further validation, we also considered radar altimetry data from G-REALM (Birkett et al., 2011) using the Elevation-Storage relationship shown in Fig. 5.C. However, the storage volume derived from G-REALM showed unrealistic behaviour in the recession and only added a few data points during the filling periods, which made us decide to exclude this source from the validation."

[Figure]

**Fig. 1:** Panel (a): dam body and G-realm measurement location (Birkett et al., 2011), Panel (b): GERD storage volume obtained using G-realm (blue) and Landsat (before improvement: orange, after improvement: green).

**References:**

Birkett, C., Reynolds, C., Beckley, B., & Doorn, B. (2011). From research to operations: The USDA global reservoir and lake monitor. Coastal altimetry, 19-50.

Melsen, L. A., Addor, N., Mizukami, N., Newman, A. J., Torfs, P. J., Clark, M. P., ... & Teuling, A. J. (2018). Mapping (dis) agreement in hydrologic projections. Hydrology and Earth System Sciences, 22(3), 1775-1791.

Vu, D. T., Dang, T. D., Galelli, S., & Hossain, F. (2022). Satellite observations reveal 13 years of reservoir filling strategies, operating rules, and hydrological alterations in the Upper Mekong River basin. Hydrology and Earth System Sciences, 26(9), 2345-2364.

**Response to referee comment: Anonymous Referee #2**

* Please note that page and line numbers in the response are based on the updated manuscript!

**Comment:** Summary: This study uses a lumped hydrological model, along with (not "coupled with" as noted in the abstract) remote sensing data to examine the filling strategies of the Grand Ethiopian Renaissance Dam (GERD). The model used is the HBC-light model, which is used to simulate the inflow into the reservoir, evaporation etc. Overall, it is an interesting study that presents substantial results on inflow, outflow, and filling strategies for GERD. The paper is generally well written and there is a lot to like in the paper, thus I am generally supportive of the work and believe that it could eventually be published in HESS; however, substantial revisions are necessary before the paper can be accepted. I provide my detailed comments below.

** Abstract, Line 6, "coupled": I don't think the model is coupled with remote sensing data. RS data is used in conjunction with the model. Please revise this statement.

**Response:** We changed "coupled with" to "in conjunction with".

**Comment:** ** Abstract and conclusion: As I noted above, there is a lot going on in this study; however, I am not convinced that the study, at least as it stands in the current form, presents sufficient novel scientific insights. It surely presents substantial information that could be used to manage reservoirs in the study region, but I ask: what is the scientific merit? I suggest that authors revise the introduction to address this issue, and perhaps some changes in the results and conclusion sections should be made as well.

** End of introduction (Lines 70-79): this is not very convincing. Again, what is the major scientific contribution of this work? Please clearly specify scientific questions and objectives. The authors attempt to justify the study (toward the end of the paper) noting that the approach/framework could be generally applicable to other (data sparse) regions; I am not sure how valid this claim is given substantial uncertainties in the ability to simulate the flows by the model and the inherent limitations in remote sensing data.

**Response:** We agree that the novelty of the work can be better clarified. Therefore, we slightly reformulate the abstract, introduction, and the conclusion to better present the novelty of the study.

We reformulated the following paragraph and added three research questions:

P3 L67: "In this study, we aim to develop a novel approach to infer reservoir management strategies under limited data availability. In our approach, two techniques, hydrological modelling and Earth observation, are combined to retrieve reliable reservoir records. Then, we reconstruct management strategies based on further analysis of the retrieved records. The merits of this approach include: (1) facilitating proactive flow forecasting for real-time operations, (2) generating refined and more consistent reservoir data, thereby supporting hydrological analyses at various temporal scales, and (3) permitting the assessment of the influence of different input uncertainties on the inference of the management strategies."

P4 L80: "1. How can hydrological modelling and remote sensing be leveraged to effectively infer GERD management strategies?

2. What is the implication of rainfall selection and parameter uncertainty on the inference of GERD management strategies?

3. What is the impact of GERD reservoir filling on the flow to Sudan during the first three phases (2020 - 2022)?"

**Comment:** ** The simulated inflow is somewhat questionable as it is not validated with any observed data. Given many sources of uncertainty, how do the authors ensure that the simulated inflow is reasonable?

**Response:** The model was validated with observed data at Eldiem station. This station is close to the dam location, hence we assume that the discharge in both locations (before dam construction) are equal. Calibration and validation were done prior to dam construction to ensure adequate representation of catchment hydrology. The calibrated model was then used to estimate dam inflow, which was proved to be reliable based on good agreement with dam storage data retrieved from Landsat and Sentinel. Our proposed approach uses three-steps validation to ensure the reliability of the results. These steps are: (1) precipitation products with rain gauges, (2) simulated discharge with streamflow gauge, and (3) inferred storage with satellite-driven storage.

**Comment:** ** Figure 4 (related to the above comment), "Best simulation": I assume "observation" here is the outflow and "Best simulation" is the inflow. Please clarify by changing the legends.

**Response:** Figure 4 and D2: "Observation" was changed to "Outflow (observation)" and "Best Simulation" to "Inflow (best simulation)"

**Comment:** ** Figure 4: Is the unit "MCM"? Discharge should have a unit with per unit time, not just volume! Same applies to the right axis of Figure 11.

**Response:**

The reviewer is correct. Figure 4, 11 and D2: the unit "MCM" was changed to "MCM d$^{-1}$"

**Comment:** ** Figure 7 and other relevant section: Given limitations in Landsat imageries, could the authors use other remote sensing products such as those from Sentinel?

**Response:** We added analysis using Sentinel imageries. Therefore, we made the following changes:

1) We added a short paragraph describing the methodology used:

P9 L206: "On the other hand, the computation of WSA from Sentinel-2 imagery was performed using the PyGEE-SWToolbox (Owusu et al., 2022). This open-source toolbox was observed to be a user-friendly, adaptable, and dependable method for quantifying WSA. The software provides a graphical user interface for obtaining WSA time-series data from diverse satellite imagery datasets, including Sentinel-2. Consequently, by utilizing NDWI calculations and a predetermined threshold value of 0, we

can accurately detect and classify pixels as water bodies (i.e., NDWI > 0). Detailed steps can be seen in Fig. A1.B."

2)  We modified Fig. 6 to include GERD area and storage estimated based on Sentinel-2 imageries:

[Figure]

**Figure 6.** Satellite-derived GERD reservoir information during the filling period 2020 - 2022. Panel (A) shows the derived water surface area while panel (B) shows the derived storage volume of GERD using Landsat (Green: before improvement; Light orange: after improvement) and Sentinel (blue). The WSA maps of the dates specified in cyan dotted lines are illustrated in Fig. 7. Note that not filling the gaps induced by the Scan Line failure of Landsat 7 produces NDWI = 0 in these gaps when using the algorithm of Vu et al. (2022). Therefore, in case of negative NDWI thresholds (which were the case here), Landsat 7 overestimates the areas and storages.

3)  Since the results of Sentinel and Landsat were almost similar, our results described Landsat only and mentioned Sentinel when necessary. Therefore, HBV-Light performance against Sentinel was added in the appendix:

[Figure]

**Figure D3.** HBV-light performance in retrieving the storage volume of GERD from 2020 to 2022. (A, B) The Nash-Sutcliffe Efficiency (NSE) and (C, D) the Root Mean Square Error (RMSE) between the satellite observed storage (using Sentinel) and the simulated GERD reservoir storage (using HBV light). Panel (B) is a zoom-in of (A) showing NSE between 0 and 1 while (D) presents the RMSE of all

simulations with NSE > 0. The x-axis is the three tested rainfall products for input uncertainty while the whiskers demonstrate HBV-light parameter uncertainty.

4) We updated Table 5 to include the results of HBV-Light best simulation against Sentinel:

**Table 5.** The performance of HBV-Light best simulation in retrieving the storage volume of GERD reservoir taking Satellite-derived storage volume as reference.

| Satellite Rainfall Product | Landsat | | Sentinel | |
|---|---|---|---|---|
| | NSE | RMSE (BCM) | NSE | RMSE (BCM) |
| CHIRPS | 0.77 | 1.70 | 0.86 | 1.52 |
| PERSIANN-CDR | −4.47 | 6.26 | −2.27 | −7.37 |
| ARC2 | −307.14 | 62.40 | −206.52 | −58.71 |

5) We updated Figure 9 (below) and D4 to include Sentinel data:

[Figure]

**Figure 9.** Temporal dynamics of inferred GERD filling strategies. The first three years of the Grand Ethiopian Renaissance Dam (GERD) filling strategies were retrieved from HBV-light lumped model using CHIRPS. The first row is the daily fraction of inflow volume retained by GERD. The second and third rows are the daily water storage and the corresponding water surface area of the GERD reservoir, respectively. The last row is the daily evaporated volume of water from the reservoir surface. The prediction uncertainty ranges indicate the 95, 60, 40, and 20 percentiles of the results obtained from the simulations classified as very good which were 1756.

6) We also added new figure (in the appendix) including the steps of Landsat- and Sentinel-driven area and storage:

[Figure]

**Figure A1.** Procedure to derive GERD reservoir characteristics using Landsat (A) and Sentinel (B). Landsat water surface area (WSA) was estimated based on the modified algorithm by Vu et al. (2022). Fixed NDWI threshold was used to estimate WSA for Sentinel images using PyGEE-SWToolbox (Owusu et al., 2022)

**Comment:** ** There are high uncertainties/errors in many of the results presented, which are not discussed in the present manuscript. The uncertainties arising from precipitation data are discussed, but there are many other sources of uncertainties including the use of PET data from ERA5. Is the PET data reliable? How do any uncertainties in the PET data affect the results. I suggest that the authors present a dedicated (concise) section on various sources of uncertainties, the implication on the results, and potentially the conclusion drawn.

**Response:**

P22 L416: We added "The retrieved GERD storage results are dependent on HBV-light performance, which is subject to various sources of uncertainty, including observed discharge, meteorological products, and model parameters and structures. Rainfall products were found to be a main source of uncertainty with a high influence on the HBV-light results, followed by model parameters (as discussed in section 4.3.3). Although uncertainty arising from potential evapotranspiration products was not investigated, previous studies (Nonki et al., 2021; Wang et al., 2022) have concluded that it is unlikely to have a significant effect on HBV-light performance and simulations. The selected model structure, HBV-light, is also likely to impact the results. However, the limited data availability limits opportunities to explore more complex structures."

**Comment:** ** L403: The question here suffers from critical grammatical issues. Please carefully rewrite it.

**Response:** P23 L453: We corrected the question to be "How is the precipitation product selected?"

**Comment:** Minor comments:

** There are excessive abbreviations used in this paper. I suggest removing those that are not necessary.

**Response:** We removed unnecessary abbreviations which are MW, HAD, CGLS, NMA, and ENTRO.

**Comment:** ** I found that figure number is not done in an increasing order in the text.

**Response:** This is mainly in L180 (in the old manuscript). Therefore, we split Fig. 6.A to be in the appendix (Fig. A1) and we added the steps of Sentinel as well. The following (appendix) figures were updated accordingly.

Also, the arrangement/placement of figures in the text is not good; many figures are 2-3 pages away from where they are referenced in the text, which makes it hard to read.

**Response:** We agree. Therefore, we improved the arrangement of the figures, but the placing is partly imposed by the HESS format.

**Comment:** ** L85: UBN was already spelled out.

**Response:** P4 L92: We replaced "the Upper Blue Nile (UBN) basin" by "the UBN basin""

**Comment:** ** L118: What does "satellite imagery" mean here? Please specify.

**Response:** P5 L125: After "… satellite imagery" we added "(Landsat and Sentinel)"

**Comment:** ** L210: "is" à "are"?

**Response:** P9 L227: We corrected "is" to "are"

**Comment:** ** L248: "shows" = "show"

**Response:** P12 L267: We corrected "shows" to "show"

**References:**

Nonki, R. M., Lenouo, A., Lennard, C. J., Tshimanga, R. M., and Tchawoua, C.: Comparison between dynamic and static sensitivity analysis approaches for impact assessment of different potential evapotranspiration methods on hydrological models' performance, Journal of Hydrometeorology, 22, 2713–2730, https://doi.org/10.1175/JHM-D-20-0192.1, 2021.

Owusu, C., Snigdha, N. J., Martin, M. T., and Kalyanapu, A. J.: PyGEE-SWToolbox: A Python Jupyter Notebook Toolbox for Interactive Surface Water Mapping and Analysis Using Google Earth Engine, Sustainability, 14, 2557, https://doi.org/10.3390/su14052557, 2022.

Seibert, J.: HBV light, User's manual, Uppsala University, Institute of Earth Science, Department of Hydrology, Uppsala, 1996.

Vu, D. T., Dang, T. D., Galelli, S., and Hossain, F.: Satellite observations reveal 13 years of reservoir filling strategies, operating rules, and hydrological alterations in the Upper Mekong River basin, Hydrology and Earth System Sciences, 26, 2345–2364, https://doi.org/10.5194/hess-26-2345-2022, 2022.

Wang, C., Si, J., Li, Z., Zhao, C., Jia, B., Celestin, S., He, X., Zhou, D., Qin, J., and Zhu, X.: Evaluation of three gridded potential evapotranspiration datasets for streamflow simulation in three inland river basins in the arid Hexi Corridor, Northwest China, Journal of Hydrology: Regional Studies, 44, 101 234, https://doi.org/10.1016/j.ejrh.2022.101234, 2022.

---

## Author Response (AR2)

**Response to Editor comment:**

Dear Authors,

The first reviewer is happy with the paper as it is. The second referee however not and ask for some additional justification of statements and methods. I therefore advice minor revisions.

Dear Editor,

We sincerely thank you and the two anonymous reviewers for dedicating your time and expertise to reviewing our revised manuscript. Your feedback and constructive comments have been valuable in improving the quality of our work.

We are grateful to reviewer #2 for expressing satisfaction with our response and the modifications we made. In response to the valuable feedback raised by reviewer #1, we are hoping that our following response is sufficient.

Best regards,

Awad M. Ali (on behalf of the authors)

**Response to referee comment: Anonymous Referee #1**

* Please note that page and line numbers in the response are based on the updated manuscript!

**Comment:** The paper contributes a reconstruction of the filling strategy of the Grand Ethiopian Renaissance Dam (GERD) in Ethiopia based on a combination of hydrological modeling and satellite data processing. The authors have improved their original submission but my major comment related to the water management implications of their results has not been adequately addressed.

In their replies and the associated revision of the manuscripts, the authors argue that "No other study has presented quantitative information about the impact of GERD on downstream discharge, which is crucial for water managers in Sudan to optimize the operation of the dams for food security and hydropower generation. With the information provided in our manuscript, it is possible, for example, to reschedule the filling of the Roseires dam to align with agricultural requirements in the coming years". And also "Moreover, hydrological modelling allows to investigate future scenarios under different management strategies". I think both these arguments are potentially questionable and would invite the authors to better elaborate on this important point.

About the first reply, while I agree about the timely contribution of quantitative analysis related to the GERD filling, it's important to notice that this analysis indeed captures the initial phase of a transient period – the filling – which is not informative on how the rest of the filling and the regime operation of the GERD will be. As a consequence, the results provide an interesting retrospective analysis but I don't think can inform a reoperation of the Roseires dam.

We do agree that our evidence not fully supports our first claim in its current form:

"With the information provided in our manuscript, it is possible, for example, to reschedule the filling of the Roseires dam to align with agricultural requirements in the coming years"

Therefore, we have edited our paragraph to include the sentences that are underlined:

P23 L432: "The operation of downstream dams (i.e., Roseires and Sennar) depends on the management of the GERD dam. Thus, the outcomes of this approach provide information that can support downstream dam operations. This study creates an opportunity, for instance, to adjust the scheduling of the Roseires dam filling to coincide with agricultural needs in the upcoming years, specifically during the filling phase. Upon examining Figure 10, one could speculate that the 4th filling might occur around July and August, encompassing approximately 40% (potentially up to 50%) of the monthly inflow. In addition, the findings of this study indicate that lower discharge levels were observed during July and August, while higher discharge rates were observed during other periods. It suggests that additional water should be stored during GERD-non-filling periods to achieve the targeted monthly elevation of the Roseires reservoir. A substantial increase in discharge is noted during March-June and October-December. We propose that effective management of Roseires and Sennar should involve storing additional water during these months to compensate for shortages during filling periods. Additionally, for adaptive management of Sudanese reservoirs, the existing model should be expanded to include the Lower Blue Nile basin and used to evaluate alternative operation scenarios for Roseires and Sennar dams. This approach will enable a comprehensive understanding of the implications of various management decisions and facilitate the development of optimized reservoir management plans. Moreover, our method has the potential to predict upcoming reservoir management by integrating our developed hydrological model with forthcoming discharge measurements, allowing for the reconstruction of operational strategies."

As for the second reply, the hydrological model developed by the authors actually reproduces the hydrology of the Blue Nile Basin without the GERD, with the comparison of the simulated natural discharge against the observed values providing information about the implemented filling strategy. How can you use this model for investigating scenarios with different management strategies if the GERD is not part of the model?

Overall, I would suggest the need for another round of revision to allow the authors to clarify these important points before accepting the paper for publication.

Concerning our second statement: "Moreover, hydrological modelling allows to investigate future scenarios under different management strategies"

In this context, we are not specifically referring to our hydrological model. What we intend to convey is that hydrological modelling in general, unlike the utilization of satellite imagery, offers the capability to simulate potential future scenarios, assess their impacts, and explore alternative management strategies. Returning to the core objective stated in lines 5-7, our study aims to introduce an innovative approach for obtaining reservoir filling information. Our findings indicate that our approach holds promise for inferring reservoir filling patterns. We accordingly acknowledge that our approach necessitates further research to validate its forecasting skilfulness (including reservoir filling and operational dynamics).

Hence, we have revised our paragraph to enhance clarity, as outlined below:

P23 L447: "It is worth mentioning the main advantages of the proposed approach. Besides the approach discussed here, the downstream analysis can also be based on satellite images using, for example, the method proposed by Vu et al. (2022). However, relying on the latter approach for real-time operation presents certain challenges. Firstly, given the current availability of free satellite data (such as Sentinel and Landsat), it is not feasible to achieve daily time steps, unlike in the case of relying on hydrological modelling. Secondly, waiting for a few days to acquire satellite data can be problematic, particularly during flood events, such as those experienced in Sudan in 2020. Moreover, hydrological modelling in general offers the capability to simulate potential future scenarios, assess their impacts, and explore alternative management strategies. However, our proposed approach relies on outflow observations, which may not always be available or accessible. As such, both satellite imagery and hydrological modelling have their respective advantages and limitations. Furthermore, our approach necessitates further research to validate its forecasting skilfulness (including reservoir filling and operational dynamics)."